# Causality between Technological Innovation and Economic Growth: Evidence from the Economies of Developing Countries

**Maha Mohamed Alsebai Mohamed** [1,2] **, Pingfeng Liu** [1,*] **and Guihua Nie** [1]

1   School of Economics, Wuhan University of Technology, Luoshi Road 122, Wuhan 430070, China; maha.mohamed@fcom.bu.edu.eg (M.M.A.M.); niegh@whut.edu.cn (G.N.)
2   Department of Economic, Faculty of Commerce, Benha University, Benha 13511, Egypt
*   Correspondence: lpf@whut.edu.cn

**Abstract:** Economic growth is a tool for measuring the development and progress of countries, and technological innovation is one of the factors affecting economic growth and contributes to the development and modernization of production methods. Therefore, technological innovation is the main driver for economic growth and human progress. Spending on innovation, research and development as well as investment in innovation supports competition and progress. Accordingly, sustainable economic growth is achieved. This ensures the preservation of resources for future generations and the achievement of economic and social growth. Moreover, a sustainable educational level of the workforce, investment in research, creation of new products, and investor access to stock markets will be ensured through the development of the public and private sectors and the improvement of people's living conditions. Our study aimed to measure the impact of technological innovation on economic growth in developing countries during the period 1990–2018. To this end, the error correction model (ECM) method has been applied. The results showed that the variables are unstable in the level and stable after taking the first difference. Co-integration was also tested using the ECM, and Granger's causality test for the direction of causation. The test results showed that an increase in technological innovation indicators (such as spending on education, number of patents for residents and non-residents, R&D expenditures, number of researchers in R&D, high-tech exports, and scientific and technical research papers.) leads to an increase in economic growth in the short term and the long-run with a long-run and two-way causal relationship between technological innovation and GDP, and short-run causation spanning from technological innovation to GDP. The study also concluded that technological innovation has a direct impact on the sustainability of a country's economic growth, which is why it is crucial to adopt strong policies that encourage international investors to allocate capital for development in developing countries and thus encourage more research and development.

**Keywords:** causality relationship; Granger causality test; technological innovation; economic growth; panel models; research and development; education; developing countries





## 1. Introduction

Is it possible to achieve economic growth in the long run? If so, what is the decisive factor for the long-term growth rate? Which economies will grow faster? What kinds of approaches should decision makers use to encourage decent living conditions? These issues were central to many who wanted growth in the 1950s and 1960s, and they have continued to revive recent interest in long-term economic performance [1]. Furthermore, with the beginning of the twentieth century, as the importance of the knowledge-based economy increased, fundamental changes and new concepts emerged. Hence, the strength of any economy is based on the extent of its technological progress, as the world is today witnessing rapid developments with the emergence of successive new technologies; the latter playing an important role in developing societies and achieving their prosperity [2,3].

Economic growth is the continuous increase in real income in the long term, and increases in income are considered economic growth. Economic development is a structural and radical change in most of the structures of the national economy, unlike growth, which focuses only on the change in the volume of goods and services obtained by the individual represented by an increase in his average income.

Hence, economic growth is an increase in the economy's ability to produce goods and services during a specified period. It refers to the long-term expansion in the productive potential of the economy to meet the needs of individuals in society. The sustainable economic growth of the country has a positive impact on the national income and the level of employment, which leads to more standards of living. There are many factors that affect economic growth: (1) The amount of physical capital: the availability of more auxiliary tools in production processes leads to more output of goods and services, and accordingly, the output of the individual, in terms of the accumulation of capital, becomes noticeable, to the extent that it was considered at one time, that physical capital is generally the only source of economic growth. For investment opportunities that were not presented before, it is possible for this society to achieve an increase in its production capacity by increasing its balance of real capital. It must reveal, sooner or later, the decrease in the return on capital according to the decrease in its marginal productivity with every increase in the quantity used in the production process. Along this line, one of the most prominent examples of this is the impact of physical capital on the economic growth of the United States. During the current century, that is, despite the significant amounts of marginal capital used in that stage of development in the American economy, the ratio of output to capital has remained proportional to the declining trend and did not deteriorate. Extremely important is that investment opportunities have expanded at the same speed as investment in capital goods. (2) Human resources are one of the most important factors leading to increased economic growth; the quantity and quality of human resources contribute directly to the economy. The quality of human resources depends on a set of characteristics, the most important of which is their ability to innovate and provide education, training, and skills. In the event of a shortage of skilled human resources, this will hinder economic growth. (3) Natural resources are among the factors affecting the economic growth of a country. Natural resources are significant and include all the natural resources that appear on the surface of the earth or within it, such as plants on land, and water resources. Natural resources within the earth include gas, oil, and minerals. Natural resources differ between countries based on their environmental and climatic conditions. (4) Social and political factors are the factors that aim to play an important role in the economic growth of countries. Traditions, customs, and beliefs constitute social factors, while government participation in policy development and implementation constitutes political factors. (5) Technological development is one of the important and influencing factors in economic growth, and includes the application of a set of productive techniques and scientific methods, and technology is defined as the nature and quality of technical tools, dependent on the use of a certain percentage of the workforce. Technology is defined as "a set of knowledge, experiences, and practices." Technology and the interrelationships between the sub-systems of work, its application, and adoption contributes to satisfying actual or expected economic and social needs [4,5].

In the same context, (6) innovation is one of the factors that affects economic growth; innovation can be defined as "the activity that produces new or significantly improved goods (products or services), processes, marketing methods, or business organizations [6]. This definition focuses on forms of innovation. It may be embodied either in a new or improved product, and it can also be defined as "the successful commercial exploitation of new ideas" and includes all scientific, technological, organizational, and financial activities that lead to the provision of everything new (or improvement) of a product or service [7,8]. Innovation also refers to "the successful exploitation of new ideas" [9]. According to (Sarvan, Atalay, 2013), innovation can be embodied in the following manifestations: creating new products or qualitative improvements in existing products; —carrying out a new

industrial process; opening a new market; developing new sources of raw materials or other new inputs; and new forms of industrial organizations [10].

There are several types of innovation, which are usually classified according to the following criteria. Classification of innovation according to the output criterion includes two types: product innovation and process innovation. Innovation is also classified according to market perception criterion, and this classification includes two basic types: continuous innovation and intermittent or discontinuous innovation. Another way innovation is classified is according to the criterion of the size of change (according to degree). According to this criterion, innovation is divided into two types: radical innovation and improvement innovation (gradual—partial). Alternatively, a production method involves the process of achieving and embodying innovation in a tangible form.

Finally, classification of innovation is according to the criteria of specialization into managerial innovation, marketing innovation, and technological innovation." According to Garcia (2014), "Technological innovation is a set of technical, industrial and commercial stages that lead to the launch of manufactured and commercial products and the use of new technical processes [11]." Figure 1 shows the types of technological innovation.

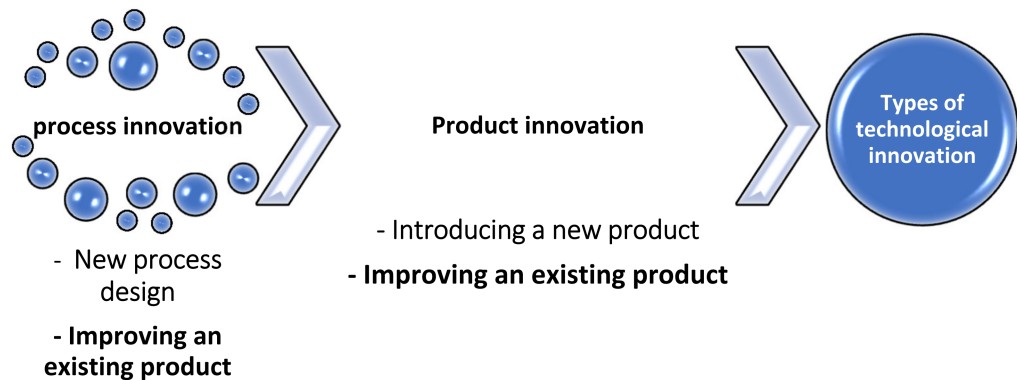

**Figure 1.** Types of technological innovation. Source: Prepared by the authors based on previous studies.

Figure 1 clearly shows that technological innovation consists of two types: product innovation, which is either introducing a new product or improving an existing product, and process innovation as the second type, which consists of designing a new process or improving an existing process. The innovative process, where countries today depend on the use of modern technology to remove many of the barriers that make the country more open and developed in terms of speed in completion of work and keeping pace with the times, by focusing on the research and development function in a way that allows it to keep pace with these developments and challenges as well as adapt to them. Countries cannot maintain their level of performance, regardless of their capabilities or capabilities, if they rely on traditional methods in the era of the technological revolution. As such, countries must rely on technological innovation, which is one of the most important pillars for the development of countries where they can reach the required level of performance efficiently and effectively.

Therefore, the pursuit of technological innovation is one of the main driving factors that make developing countries more advanced and ambitious. Sustainable development is a new concept that has emerged on the ground, is concerned with preserving resources for future generations and achieving economic and social growth. Moreover, it is also concerned with preserving the environment. In addition, long-term sustainable economic development is one of the most important goals for each country. Thus, the state can achieve this goal by increasing its production. There are two ways to increase GDP: (1) by increasing the production components that we use in the manufacturing process and (2) by raising the efficiency of the inputs. This could be by improving productivity by producing innovative goods or by introducing new manufacturing methods.

In the context of developing countries, which have gone through a transition from agricultural to industrial societies, whose economies do not focus on knowledge (creativity and dissemination) and use of science and technology compared to developed countries, whose quality of life is lower, the human development index (HDI) and per capita income are relatively low. While the main share of data production (innovation) takes place in developed countries, innovations in the north "strongly believe in radical development. This does not undermine the importance of innovation (and studies analyzing its processes) in developing countries, although innovation in developing countries does not contribute significantly to the frontiers of global knowledge; at least its impact should be vital and effective in the developing country and increase per capita national income [12].

In this respect, technological innovation has played a leading part in economic growth, creating innovative energy opportunities. One of the positive effects of technological innovation is the diversification of energy sources simultaneously and with the same devices, which contributes to reducing pollution. In addition to producing similar alternatives from more effective materials at the cheapest cost and with less pollution, this contributes to the increase in flexibility of the production system and the reduction of production costs. Moreover, the marketing of modern technologies leads to increased accuracy in production by adhering to the specified standards and specifications according to scientific principles that are not harmful to the environment. In addition, maintaining the latent reserves of renewable materials contributes to maintaining the ecological integrity of these resources. It can be said in another way that the innovation of technologies with scientific specifications works to preserve the environment by avoiding environmental pollution to its surroundings. In addition, upgrading societal prosperity aside from technological alterations has been consistent with most artistic research for a long time. Interestingly, invention creates opportunities in developed countries as much so as in less developed countries [13,14]. Therefore, technological innovation may take three forms: cost savings, quality improvements, or expansion in a variety of products, services, and manufacturing methods. Innovation is finding new and better ways to conduct business and bringing new ideas or new types of products and services to the market [15]. Therefore, innovation is carrying out new things in a new way. Innovation transforms and develops the technological qualities and performance characteristics of goods and processes, and changes organizational forms and market strategies, thereby adding dynamic change and efficiency development to the financial system. To try anything different, companies need to learn—if they do not learn, nothing new happens [16–18]. As is the case for many developing countries, foreign R&D is a vital technology resource; the share of domestic R&D in Egypt's GDP is 0.7%, of which only 8% is undertaken by the business sector [19].

As well, some researchers are trying to focus on a particular aspect or process because of the intricacy of the innovation operation. For instance, Porter connects innovation with competition, and Cooper links innovation with spread; other scholars attribute creativity to practice and preparedness. Romer also shows that the use of a larger variety of inputs in output (new products and intermediate goods) enables per capita production to be increased. The model of innovation embodies the idea of horizontal imagination (which is the sum of intermediate varieties added to production). According to Romeo, growth in per capita income is strongly proportional to a country's researchers' output, a remarkable finding [20–22]. Furthermore, Aghion et al. emphasize what has been rooted in Schumpeter, referring to the concept of "creative destruction", according to which innovations replace outdated products and technologies, which has a positive effect on the evolution of the growth rate. Therefore, competition in the market resulting from the creation of new innovations and the exclusion of old technologies supports economic competitiveness and promotes and sustains economic growth [23].

The significance of our study is that since the second half of the twentieth century, the world has moved towards expanding the use of the internet and its applications with continuous progress in practical research and breakthroughs in the world of innovation and technology have been witnessed, which have been reflected positively and gradually

in the sectors of the economy, industry, health, and agriculture. Researchers addressed this phenomenon through observation and analysis, and some described it as the fourth industrial revolution. In this context, the importance of research emerges: it "leads innovation to economic growth and thus the economies of developing countries through technology and innovation", to advanced economies and therefore developing countries to study the strategic importance of innovation and technology in providing unconventional solutions to global challenges, especially with increasing demand for food, energy, and water, and how to promote global urbanization. It then attempts to redraw the mental image of rich countries as not monopolizing the capabilities of innovation, but it needs the forces of minds from all over the world to provide solutions to common global challenges and to exchange and distribute international burdens.

Hence, economic progress is no longer associated with the possession of natural resources or material possibilities "as it is linked to the content of knowledge 'technology' quality and innovation". Japan is a country without resources, but with attention to human resources and economic innovation, Japan was able to be among the most important economies in the world and achieved the highest rates of gross domestic product. Thus, the research problem lies in the fact that countries suffering from weak knowledge and technology content cannot upgrade. The economy and economic growth may be unlike developed countries that possess advanced technology and have a long history of innovation, nevertheless, a country can achieve great economic growth. Growth in developing countries faces serious constraints, partly due to the lack of innovation, which is at the same time the reason for these countries remaining underdeveloped. These barriers arise from inappropriate business activity, governance, and poor education. In such cases, innovation itself is encouraged to deal with difficult situations [24].

The following is an explanation of the importance of this study:

- It is considered an important applied research design in studying the relationship between technological innovation and economic growth.
- It intensifies the research and development process for the purpose of changing the traditional structures in developing countries, and thus provides new goods that would improve the financial conditions and consequently the economic growth of countries.
- It highlights the transition from that of the traditional economy to the innovative one, by acquiring various skills that enable countries to improve their financial performance.
- The researcher expects, through this study, to motivate researchers to conduct more research in the field of technological innovation, especially on the relationship between innovation, sustainable development, and competitiveness.

Based on the foregoing factors, the study problem is limited to answering the following research question:

Is there a causal relationship between economic growth and technological innovation in the group of developing countries under study?

Our study is based on the main hypothesis that:

- There is a long-term causal relationship between economic growth and technological innovation in both directions for the group of developing countries under study.
- Our study contributes to the literature in the following ways. First, to our knowledge, this is the first study to find a systematic relationship between technological innovation and growth in the economy of developing countries. There are many studies that discuss the relationship between innovation in general and economic growth; our study focused on technological innovation because it is considered one of the most important types of innovation in addition to being one of the basic and important activities of contemporary institutions, as the main reason for the existence of institutions is to provide distinguished products and services. In order for it to survive and grow, it must adapt to changes in the external environment and find the necessary methods and processes to enable it to offer all new or improved products and services to achieve superiority over competitors, especially in the developing economy. Second,

we document this through the results of the study, where the technological innovation index represented by the percentage of spending on education is generally expected to have a positive impact on countries, our results were completely different. We found a negative and moral impact of 1% and this does not fit with the various theoretical analyses that considered spending on education as a driver of economic growth. This is because developing countries still need to spend more on education infrastructure for innovation to deliver its expected results.

The remainder of this paper is organized as follows. The Theoretical Background follows in Section 2. In Section 3, the materials description, and variables used in the analysis are included in the Research Model; causality tests were performed at Granger, followed by a co-integration and error correction model (ECM). Section 4 presents the Estimating and Analyzing Results. In Section 5, implications of findings are presented in the Discussion. Finally, Conclusions in Section 6.

## 2. Theoretical Background

There is a substantial amount of empirical literature focusing on technological innovation and economic growth that has consistently shown that technological innovation is a critical catalyst in economic growth. Among the most important studies focusing on this aspect is the research of Freimane et al. who used research and development as a measure of innovative activities [25–29]. The economist Joseph Schumpeter considered that innovation is one of the productive functions and emphasized that entrepreneurs are able to achieve these innovations, and thus, entrepreneurship plays a fundamental role in economic growth [30]. Theoretically, the innovation-based growth hypothesis suggests that there is a positive linkage between innovation and economic growth. According to this hypothesis, R&D plays a major role in innovation, raising productivity, and accelerating economic growth [31–33]. Based on existing literature, this paper systematically sorts research related to the relationship of innovation technologies to economic growth from aspects of semantics and characteristics, composition and development, innovation, and management of emerging technologies. Various theories explain the relationship between technology innovation and economic growth. In the neoclassical context, the impact of innovation is seen as part of the Solow residual and thus a major contributor to economic growth and long-term integration [34]. The Solow residual is a number describing empirical productivity growth in an economy from year to year, and decade to decade. Robert Solow, the Nobel Memorial Prize in Economic Sciences-winning economist, defined rising productivity as rising output with constant capital and labor input. It is a "residual" because it is the part of the growth that is not accounted for by measures of capital accumulation or increased labor input. Increased physical throughput, i.e., environmental resources, is specifically excluded from the calculation; thus, some portion of the residual can be ascribed to increased physical throughput. The example used is for the intra-capital substitution of aluminum fixtures for steel during which the inputs do not alter. This differs in almost every other economic circumstance in which there are many other variables. According to the "Solow surplus" model, the unexplained portion of economic growth, except labor and capital increase, is technological development. The convergence hypothesis, which is one of the main implications of the Solow model, is based on the assumption that technological change is external and constant between countries. Accordingly, per capita output levels of countries will approach each other, and the development differences will automatically disappear in the long term [35].

Technological change is one of the most important challenges facing countries for its strategic role in achieving outstanding performance, maintaining its competitive advantage in the markets, and its sustainability, survival, and success in the fields of work. Technological change is a more comprehensive concept than development, growth, and progress. Technological change is what leads to development; technological development can be defined as a set of activities related to examining, evaluating, and implementing an idea or goal for the purpose of moving from the research mental level to the production level, and

includes developing processes for technical capabilities, performance, design, engineering models, and manufacturability. While technological growth means a continuous increase in technology over time, technological progress is the change in the art of production used, leading to an increase in productivity, provided that the ratio of capital and labor use remains constant.

Both Ricardo and Adam Smith emphasize that openness will enhance specialization and thus countries will specialize in the production of goods and services that have advantages and export these goods and services; on the other hand, countries that do not have these advantages will import from those countries and specialize in other types of goods and services, and as a result, resources are allocated optimally. The theory of internal growth indicates that developing countries will benefit from the transfer of advanced technology through a policy of trade openness, this technology can be exploited in productive processes and thus achieve a large production that is directly reflected in economic growth [36].

The neoclassical growth models derived from Solow's 1957 model consider a technological change to be exogenous and suggest that trade policies do not, therefore, affect economic growth. However, new economic growth theories assume that technological change is an endogenous variable [37].

Thus, modern growth theories have emerged, which are termed internal growth theories, with the contributions of Romer and Lucas, and the theory of internal growth focused on the internal impact of technological change, research and development, human capital, and their impact on the production function [38,39]. In-house designed technological change generates sustainable economic growth, assuming constant returns to innovative research, in terms of human capital used in research and development (R&D). Internal growth models provide an appropriate framework for examining important issues related to the role of technological change in the process of economic growth, as well as design, research and development efficiency and innovation policies. "Barro" focused on infrastructure and public expenditures, and others have focused on economic openness and its role in economic growth [40].

Paul Romer's model of endogenous growth distinguishes between inputs and outputs. His knowledge takes the form of a number of ideas (designs) that are embodied in the form of a number of (technical) inputs, which in turn are embodied in the form of final goods and services. Hence, Romer's model links the sector of the production of ideas and designs (research and development), the sector of input production (the sector of production of intermediate goods), the sector of capital production (which is just a mixture of inputs) and the sector of production of goods and services [22,41]. Hence, it can be said—according to Romer's model—that designs constitute the output of the knowledge economy, while the inputs that are used in the production of capital and in the final goods production sector represent the impact of the knowledge economy on the knowledge-based economy. Thus, this relationship between these sectors is logical to govern—in principle—the logic of designing and building knowledge standards, knowledge economy standards, and knowledge-based economy standards. Romer concludes that growth is often driven by the accumulation of non-competitive inputs (intermediate inputs), but they are partially enumerated, and by competitive inputs, are embodied in human capital, not by the size of the labor force or the size of the population [22]. Thus, the transition from a product economy to a knowledge economy has some consequences, including providing an opportunity to increase returns, such as what happened in the industries software sector, as well as creating the opportunity to benefit freely, by taking advantage of knowledge outputs [42].

In the same context, some studies, including Aghion and Howitt, Chu, and Jinli Zeng, indicate that capital accumulation (both physical and human) and innovation should not be considered as causal factors differentiate, but are manifestations of a single process. On the one hand, capital is used in the innovation process and in new technology applications resulting from research and development activities. Hence, long-term growth depends on

both capital accumulation and innovation. On the other hand, new technologies create new economic opportunities for investment in physical and human capital [43–45]. Nelson has indicated that knowledge takes the first priority compared to the traditional factors of production, material, and financial. Unlike land, labor, and capital, which were highlighted by traditional economists as final factors of production, knowledge, and ideas are infinite goods and help to obtain increased benefits; the new economists link the theory of superior growth creativity emanating through the system [46]. Nelson also emphasized that the level of innovative activity in a country is determined by the level of interaction of specialized [47] institutions among them [48]. Hence, a review of these different theories confirms that technological progress appears in them as a supportive factor for productivity growth and thus achieving long-term economic growth [49]. Expenditure on scientific research, technical development, education, and rehabilitation of human capital is one of the most important tools supporting innovation [39].

Hence, most innovation studies are focused on developing solutions to technology problems. Researchers have tried to show how the organization can develop technological solutions to the problems they face, where technology is seen as solutions to problems [50]. In addition, the results of much quantitative research confirm that the development of technological capabilities is a prerequisite for reducing the difference in economic development between countries and thus achieving the so-called catch-up growth in developed countries [51]. This means reducing the difference in the level of income per capita. Many countries, such as Japan, South Korea, and others have also achieved this. The economist Kim interpreted the economic development in South Korea on the basis of the development of its technological capabilities, which is known as the ability to effectively use technical knowledge to imitate, invest, localize, and modify the existing technology. Technology capabilities are also a necessary condition for achieving technology transfer and settlement [52], whereas innovation potential describes a country's ability to produce and market innovative technology over the long term [53]. The financial and scientific resources necessary for innovation and the results of scientific research are the most important factors that affect the innovative potential of a country [54]. Furthermore, human capital, infrastructure, and foreign trade are among the most important factors affecting this country's ability to absorb new technology, achieving development based on innovation and thus achieving economic growth.

Based on the foregoing studies, technological change can be defined as "the use of innovation or creativity outputs for the purpose of bringing about a partial or total change in the production process, or the product that aims to support competitiveness and therefore continuous modification in it to achieve continuity and growth". It is often claimed that the impact of progress on economic development cannot be fully appreciated without considering the social and structural structures of the country. For example, Rodriguez and Crescenzi demonstrated how the interaction between research and socio-economic and institutional conditions shapes the potential for regional innovation [55].

Tuna et al. focused on analyzing the relationship between research and development (R&D) expenditures and economic growth in Turkey, using unit root tests, the concurrent integration test, and Granger's causation. The results of the analysis showed that the time series are stable in the first degree, and there is no simultaneous integration relationship between them. According to Granger's causal analysis, it was revealed that there is no causal relationship between the tested time series [56].

Abdelaoui. et al., aimed to measure the impact of innovation on economic development in Algeria, Tunisia, Morocco, Egypt, the United Arab Emirates, Kuwait, and Saudi Arabia, for the period 2007–2016. It lists several composite indicators that go beyond traditional measures of innovation, such as research and development expenditures and the number of trademarks and patents. The impact of innovation has been measured on the following independent variables: the growth of per capita real output, the unemployment rate, and the human development index, as indicators that measure levels of economic development. The economic measurement of the panel data was used based

on the apparently unrelated equations method and the middle of the combined group method. The study concluded that there is a significant positive impact of innovation on the growth of per capita output as well as unemployment, and the results indicated the role of innovation in improving human development levels [57]. Lomachynska and Podgorna examined the causal relationship between innovation, financial development, and economic growth using panel VAR modeling for a sample of 27 OECD countries during the period 2001–2016. The adopted approach allows downloading the triple links between innovation, financial development, and economic growth. The study concluded that there is a one-way causality from economic growth to financial development. The results of the study also confirm the hypothesis of neutrality from financial development to economic growth, as well as between innovation and economic growth and between financial development and innovation [58]. Pece et al., examined whether long-term economic growth is affected by innovation potential through the use of multiple regression models estimated for central European countries using the following measures: with regard to economic growth and patents, a number of trade currencies, research, and development expenditures for innovation, and by using regression models to estimate the relationship between economic growth, investment, and innovation, the results represent a strong relationship between humans, money, and economic growth [59]. Solomon et al., aimed to analyze the dependencies between growth and volatility (the degree of variance in the trading price series over time, measured by the standard deviation of logarithmic returns) and innovation in the case of the European Union and the two new member countries, and the length of the extension. The multi-regression model used the variable GDP for economic growth, and the innovation index for innovation, the regressions of the GDP growth rate were estimated on its total volatility as well as its partial volatility divided by the variables of the rate of growth related to the role of innovation. The most important results were the following: there is a positive and moral partial correlation between GDP and innovation and there is a positive and moral partial correlation between GDP growth and its fluctuations between stages [60].

Whereas the neoclassical economy recognizes that technological innovation is critical for economic growth and considers the internal technical innovation variable to be external, it may not distinguish the roots of technological development, and exaggerates economic growth as a promotional base for technological innovation.

The current theory of economic growth states that technological progress is affected primarily by considerations such as the allocation of human capital and the number of intellectual services, and identifies different models of research that depend on sophistication and scope. Since the theory of economic growth has undergone a long cycle of development, it has many methods of analysis. However, there is already agreement on the fact that technological innovation is the driving factor behind the progress of economic growth and economic development linked as cause and effect, stimulating, and assimilating each other and often forming a partnership between the two. In other words, technology innovation and economic growth overlap, both of which change in the same direction at the same time, and this relationship shows an enhanced role for technology innovation in economic growth. Growth economists, development economists, and economic historians all seem to agree on the importance of technological innovation for long-term economic growth. Even a recent article in The Economist entitled "Economists understand little about the causes of growth" nonetheless acknowledged that "growth is fundamentally about using technologies to become more productive and uncover new ideas" [61]. Several studies have analyzed the impact of technology innovation (research and development, high-tech exports) on economic growth. According to Maradana and others, the relationship between innovation and economic growth has recently emerged as a major study subject [25]. Research on this topic can be classified into four categories: Supply-leading Hypothesis, Demand-following Hypotheses, The Feedback Hypothesis, and the Neutrality Hypothesis, they are as follows:

- Supply-leading Hypothesis (SLH) suggests a unidirectional causality between innovation activities and economic growth (see, for example, Yang, [62]; Guloglu and Tekin, [40]; Cetin, [31]; Pradhan et al.) [63].
- Demand-following Hypotheses (DFH) suggest unidirectional causality from economic growth to innovation activities (see, for example, Sinha, [64]; Cetin, [31]; Sadraoui et al., [65]; Pradhan et al., [63].
- The Feedback Hypothesis (FBH) suggests a bidirectional causality between economic growth and innovation practices (see, for example, Guloglu and Tekin, [40]; Cetin, [31]; Pradhan et al. [63].
- The Neutrality Hypothesis (NLH) suggests no association between economic growth and innovation activities (see, for example, Cetin, [31]; Pradhan et al., [63].
- Throughout history, nations seeking a successful future have relied on the discovery of the next "great idea," which often follows the accidental discovery of a great idea that propelled the country forward. Nevertheless, for the country to succeed in competition, and for its growth to continue in the current and ever-changing business environment, it must learn how to develop a thriving innovation culture—that is, a continuous ability to generate, accept, and implement creative ideas—within the country, as can be seen in Figure 2.

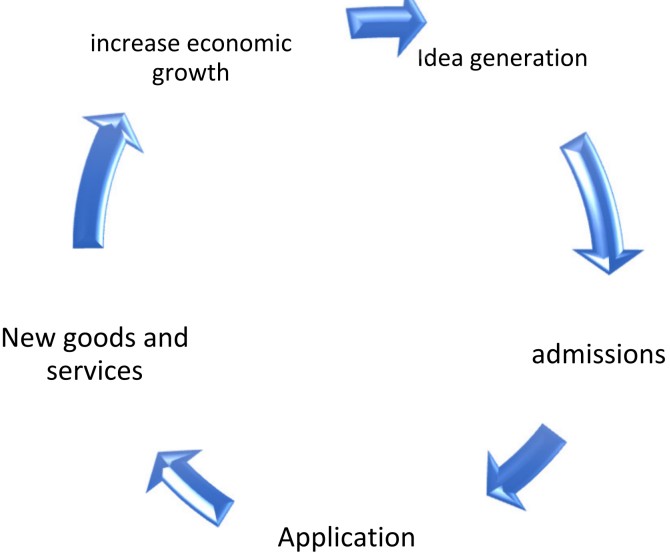

**Figure 2.** Innovation process. Source: Prepared by the authors.

The innovation process begins by generating creative ideas by finding many ideas, then choosing those that address the current problem/problems, or that make the best use of opportunities to meet the needs of the state; then comes the stage of accepting ideas that help introduce a new product or introducing a new method of production. Hence, innovation is the process of transforming new ideas and new knowledge into new products and services, and thus this activity entails opening new markets, or finding appropriate sources of raw materials. Thus, innovation is the introduction of innovation into a country's national economy, which positively affects economic growth.

From this perspective, the authors defined the main goal of the study. Hence, the article investigates to determine the direction of the relationship between technological innovation and economic growth in developing countries. Technological innovation activities are mainly seen as being hampered by a lack of funding for technological innovation, and the idea of common sense has not been helped by technology and economic development being closely related [66]. This is because technology fundamentally looks for freely available knowledge that could be used without depleting it. That is why, while it may help the whole world to the same degree, it cannot justify inequalities in growth [67].

## 3. Research Model

### 3.1. Economic Methodology

Aghion and Howitt proposed a model for the variables that affect innovation: the technical multiplier, total employment committed to innovation, intermediate product output, and volume of final and medium goods produced. We demonstrate that the driving factor behind economic growth is the creation of technology developments. This outcome relates to the financial framework, that is, the demand that allows the creator to finance and, in some way, the possibility of excluding the enterprise from the business. Therefore, when referring to innovation as a transformation in the production process, as formulated by Schumpeter, Cobb-Douglas will be called the product advantage with constant returns to scale, namely [68,69]:

$$Yt = AK\,t\,\alpha\,L\,t\,1 - \alpha \tag{1}$$

where $Yt$ is production, $Kt$ is capital, $Lt$ is labor, $A$ is the technological coefficient, and $\alpha$ and $1 - \alpha$ is, respectively, The share of capital and labor in production. To assess the evolution of total factor productivity TFP, [70] should be considered, taking into account the contribution of capital to increasing demand, calculated by the change in the percentage of capital that doubles its market share. Following the same theory, the proportional change due to labor is the rise in the amount of production compounded by the share. The growth rate of the TFP is defined by variables other than labor and resources. Such considerations include the effective utilization of energy, technical developments, and innovation in R&D, patents, and exports of high-tech goods. As is normal, the TPF is obtained by taking logarithms in (1), and is given as follows:

$$TFP = gQ - SK\,gK - SLgL \tag{2}$$

where the growth rate of production is $GQ$, $SK$ is the share of capital in the industry, $gK$ is the rate of growth of capital, $SL$ is the share of labor in the product, and $gL$ is the rate of growth of labor.

### 3.2. Standard Methodology

In standard studies, panel data models refer to multi-directional data mostly that include measurements over time and that contain data for multiple phenomena over time and for the same economic units. Panel data models have become increasingly popular in the field of applied studies due to their high ability to study human behavior compared to time-series models or cross-section models, and panel data has become increasingly rich and available in all developed and developing countries alike. The World Bank is tasked with helping to design many of the surveys for panel data. Panel data modeling is carried out by adding a sample of a particular unit over time to other units within a group, thus providing multiple observations for each unit of the sample.

Panel data has several characteristics that distinguish it from data for cross-sections or time series. It works to control individual variance that may lead to biased results. It also provides an expansion of the sample size used by researchers, increasing degrees of freedom and reducing interdependence between explanatory variables, thus helping to improve the efficiency of estimates on statistical as well as cross-sectional data. On the other hand, the panel data allows researchers to analyze a number of important economic questions that cannot be studied using time series or cross-sections alone. Moreover, the panel data allows for the construction or testing of more complex models; this is carried out by utilizing information at the temporal dynamic level and at the individual level of the panels being studied. It also provides the possibility of generating more accurate predictions for units. In general, the regression can be represented as follows [71,72]:

$$Y_{it} = \alpha + \beta^1 \, X_{it} U_{it} \tag{3}$$

where as:

($Y_{it} X_{it}$) Study variable vectors.
Panel data cross-sectional directions ($i = 1, \ldots, N$),
(N) represents the number of units (people, companies, industries, countries... etc.),
($t = 1, \ldots, T$) time direction.

Stability tests: Time series are divided according to the stability characteristic into stationary series, which are series whose levels change with time without changing the average in them during a relatively long period of time, i.e., where there is no general trend towards either an increase or decrease (does not contain a unit root).

Un-stationary series are the series whose mean is constantly changing, increasing or decreasing (containing a unit root) [73], using panel data, which is defined as cross-sectional data, and are measured at certain time intervals. The main benefit of using them is to increase prediction accuracy by increasing the number of views by associating the number of cross-sectional views with the number of time periods [74].

Economic data are often characterized by the presence of structural changes that affect the degree of the indifference of time series, so determining the degree of inactivity is important before testing integration and causation relationships, as this requires data instability and integration of the same degree. If the variable is not fixed at the level, while it was fixed at the level of the first differences, then it is an integrated variable of the first degree. It is a stable time series (static) if it has the following characteristics:

- The stability of their average values over time, i.e., $E(x_t) = \mu$
- The stability of the variability of their values over time, i.e., $V(X_t) = E(X_t - \mu)2 = \sigma2$
- The covariance between two values of the same variable depends on the time gap between the two values and not on the actual value of time i.e., cove (x t, $x_t + k$) = E [($X_t - \mu$). ($X_t - k - \mu$)] = $\gamma K$

That is, the time series is considered to have a stable covariance if its means and covariances are constant [75] over time to determine if the variables ($Y_i$) are stable or not, the augmented Dickey-Fuller (1981) ADF test is performed. To perform the ADF test, the following equation is used:

$$\Delta Y_t = \alpha_0 + \beta T + \delta Y_{t-1} + \sum_{j-1}^{m-1} \beta_j \Delta Y_{t-1} + \varepsilon_t \qquad (4)$$

The instability hypothesis is rejected when the parameter is negative ($\delta$) and significant. If the variables are stable and integrated of the first degree, we move to the next step, to find out whether the variables are jointly integrated and that there is a long-term equilibrium relationship between the variables. After that, the following two hypotheses are tested:

(The variable $Y$ does not remain stationary = contains a unit root) Ho: $\beta < 0$
(The variable $Y$ rests at its level = integral of degree zero) Ha: $\beta = 0$

The null hypothesis is rejected if the calculated t value is greater than the tabulated or critical value of t (in absolute value suggested by MacKinnon (Mackinnon 1991). Nevertheless, if the variable is not static at the level while it is static at the level of the first differences, then it is an integrated variable of the first degree (1). In general, the series xt is integrated from the degree d if it is static at the level of the differences d, so it contains several d is a unit root [76–78]. After conducting unit root tests for the variables under study, it is proven that the variables are characterized as integral of the first degree (1), it is possible to conduct joint integration between them. The basis of the co-integration method is that two or more non-static variables can be co-integrated (they have a long-run equilibrium relationship) if one of them is in regression over the other and the residuals themselves are stationary. As Engle and Granger point out, even if the time series (individually) are not stationary, their linear structures can be stationary, because the equilibrium forces tend to hold these time series together in the long run. When this happens, the variables can be considered co-integrated. Hence, the error-correcting vocabulary is created to consider the short-term deviation from the long-run equilibrium relationship resulting from the co-integration [79–81].

Granger's causality test: Granger has demonstrated how to introduce the traditional method for causation testing when using the error correction model (ECM). By using, the error correction model derived from the cross-integration. The ECM also makes it possible to distinguish between the long run and the short run. Where the F and T-tests of the first difference variables' deceleration indicate causation in the short term, while the error correction factor indicates causation in the long run [82,83].

$$\Delta X_t = \sum_{j=1}^{n} a_j \Delta X_{i-j} + \sum_{i=1}^{m} a_i \Delta Y_{t-1} + \rho_1 e_{t-1} + U_1 \tag{5}$$

$$\Delta Y_t = \sum_{j=1}^{n} \beta_j \Delta Y_{i-j} + \sum_{i=1}^{m} \beta_i \Delta Y_{t-1} + \rho_2 e_{t-1} + V_1 \tag{6}$$

where: $\Delta$ is the first difference, $e_{t-1}$ error correction limit If the estimates of the two parameters $(\rho_1, \rho_2)$ are statistically significant, then this indicates the existence of a long-term causal relationship in two directions from $Y_t$ to $X_t$ and vice versa. If only $\rho_2$ is significant, then this means that there is a one-way causal relationship from $X_t$ to $Y_t$ (this implies that $X_t$ leads $Y_t$ to long-run equilibrium).

The hysteresis values $\Delta Y_t$ and $\Delta X_t$ represent explanatory variables in the model, and indicate the causal relationship in the short term. If the parameters of $\Delta Y_t$ are the previous equation number 5 is significant, it means that $Y$ causes $X$ [84].

On the other hand, if any integrative vector is not reached for a long-term relationship between the study variables, we can detect the causal relationship between variables in the short term through the Granger speaker in the multiple frame in the self-region (VAR) model:

$$Y_t = \sum_{t=1}^{m} \lambda_i Y_{i-j} + \sum_{t=1}^{m} \mu_i X_{t-1} + y + u_1 \tag{7}$$

Since: $Y_{t-1}$,1 -$X_{t-1}$ are the slowed study variables, $m$ the lag period, $\mu_i$, $\lambda_i$ are the parameters of the slowed variables $Y_t$, the random limit.

Thus, the Granger causality test is used to ascertain the extent to which there is feedback or an interrelationship between two variables. In this study, a model of the causal relationship between technological innovation and economic growth will be estimated using the Granger method.

This research seeks to test the relationship between technological innovation and economic growth achieved by the following countries: Argentina, Algeria, Brazil, Bulgaria, Chile, China, Egypt, India, Indonesia, Iran, Mexico, Morocco, Peru, Philippines, Poland, Romania, Sri Lanka, Thailand, Tunisia, and Turkey. The research was conducted during the period 1990–2018. The main variables used to measure innovation were expenditure on education (DUE): the percent of education expenditure in GDP for each of the developing countries. Public expenditure on education as a percentage of total government expenditure is the total public expenditure (current and capital) on education, expressed as a percentage of GDP in any year. Public expenditure on education includes items of government expenditure on educational institutions (public and private), education administration as well as transfers/subsidies to private entities (students/families, other private entities). Patent fields by residents (PAR), (patent applications are worldwide patent applications submitted by the mechanism of the Patent Partnership Treaties or through the Regional Patent and Trademark Office Special Innovation Protection Database, a device or method that offers an innovative way of doing things or offers an alternate technological remedy for dragging. The patent shall safeguard the innovation for a fixed amount of time, usually 20 years, from the holders of the patent (expressed in percentages and per 1000 population). Patent fields by non-residents (PAN), Researchers Scholars in research and development activities (RDE): calculated per 1000 population; patent applications are worldwide patent applications filed through the Patent Cooperation Treaty (PCT) or with a national patent office to register exclusive ownership of innovation—whether it is a product or a process

that involves a new way of making something or offers a new technical solution to a problem. A patent provides protection for the invention for the benefit of the patent owner for a limited period, generally up to 20 years. Researchers' development and expenditure (PRD): measured as a proportion of actual GDP Gross domestic expenditures on research and development (R&D), expressed as a percent of GDP. They include both capital and current expenditures in the four main sectors: business enterprise, government, higher education, and private non-profit. R&D covers basic research, applied research, and experimental development. High technology exports (HTE): measured as a proportion of the real domestic output. These are products with high R&D intensity, such as aerospace, computers, pharmaceuticals, scientific instruments, and electrical machinery. Scientific and technical journals (STG): measured per one thousand people, STGs refer to the number of scientific and engineering articles published in the following fields: physics, biology, chemistry, mathematics, clinical medicine, biomedical research, engineering and technology, and earth and space sciences. These are independent variables. In addition, GDP per capita the growth of the country's economy is measured as a percentage increase in the gross national product per capita) is an a dependent variable.

### 3.3. Availability of Data and Material

The data sets used and/or analyzed during the current study are available on World Bank Statistics.

The initial data were collected and processed using Excel 2016. Regression analysis was performed through EViews 9 the Granger causality test and GMM calculations were performed using EViews 9.

The linear Granger causality test is used to study the complex relationship between per capita GDP growth (first variations in per capita GDP growth), and innovation variables (first differences of each of EDUE, PAR, PAN, RDE, RRD, HTE, and STJ). The linear Granger causality test looks at Granger-cause innovation GDP per capita growth. In other words, it determines whether the GDP per capita growth at time t, and in the country, i is related to past lags of innovation, conditional on past GDP per capita growth, or whether GDP per capita growth Granger-cause innovation variables or not. More specifically, the test estimates the following regression model.

$$\Delta GDP_{it} = \alpha_0 + \sum_{K=1}^{P} \beta_{1K} \Delta GDP_{it-K} + \sum_{K=1}^{P} \beta_{2K} INN_{it-K} + \varepsilon_{it}, \tag{8}$$

where $GDP_{it}$ is the GDP per capita at time t, and in country i, $INN_{it}$ innovation at time t and country i and $\alpha_0$, $\beta_{1K}$ and $\beta_{2K}$ are regression parameters. The errors term $\varepsilon_{it}$ are assumed normally distributed and independent. For the null hypothesis, the Granger causality F-statistic is used to test the null hypothesis that the lagged coefficients of INN it is equal to zero "Innovation does not Granger-cause GDP per capita growth". Conversely, $GDP_{it}$ is the explained variable to test "GDP per capita growth does not Granger-cause innovation variables".

Analysis of data by panels has many benefits over cross-sectional or time series data analysis, Significant advantages include: (i) specific variability should be considered in making the subject-specific variables. (ii) It usually includes more descriptive, less collinear, and more independence than cross-sectional data analysis that can be presented as a T = 1 panel or time series data analysis panel with = 1, thereby increasing the efficiency of econometric estimates [85].

The general modeling method (GMM) for the study of panel data can be set out as follows:

$$Y_{it} = X_{it}\,\beta + Z_i b_i + \varepsilon_{it} \tag{9}$$

where $Y_{it}$ represents results, $X_{it}$ A matrix of explanatory variables that does not include a constant term. The heterogeneity impact is captured by the term $Z_i b_i$ This involves all measurable effects (specific transversal effect and time-specific effect), $\varepsilon_{it}$ is the error

term that incorporates residuals from differences in both cross-section and time series, the subscript i represents cross-sectional units and the Conexant t represents data points in time series.

## 4. Estimating and Analyzing Results

Based on the discussion on defining the direction of the relationship between technology innovation and economic growth, this paper attempts to resolve this debate using three different methodologies. Experimental methodologies differ in their statistical capabilities depending on the adaptation variables included in the models, and, accordingly, the use of different methods is useful in comparing results. Below is a description of each of these experimental methods. Statistics for minimum, maximum and standard deviations of innovation variables, PAR, PAN, RDE, RRD, HTE, STJ, and EDUE over the period (1990–2018) are presented in Table 1.

**Table 1.** Summary statistics of innovation variables, EDUE, PAR, PAN, RDE, RRD, HTE, STJ, and GDP_PER_CAPITA_GROWT.

| Indices | PAR | PAN | EDU | RDE | PRD | STJ | THE | GDP_PER_CAP-ITA_GROWT |
|---|---|---|---|---|---|---|---|---|
| Mean | 42.79799 | 55.28742 | 0.135374 | 0.022152 | 45.21019 | 153.5121 | 0.282907 | 2.974121 |
| Median | 15.97560 | 21.55963 | 0.097227 | 0.007254 | 9.439463 | 73.27613 | 0.104898 | 3.253375 |
| Maximum | 1232.796 | 1524.110 | 1.269957 | 0.533769 | 660.2663 | 1966.061 | 2.879151 | 13.63634 |
| Minimum | 0.227269 | 0.905273 | 0.000000 | $4.14 \times 10^{-5}$ | 0.281453 | 0.689034 | 0.001584 | −14.35055 |
| Std. Dev. | 85.36316 | 120.7277 | 0.157449 | 0.041215 | 103.6293 | 206.7747 | 0.483565 | 3.840783 |
| Skewness | 7.890280 | 7.811967 | 2.788900 | 5.492087 | 3.643645 | 2.860077 | 3.270931 | −0.767087 |
| Kurtosis | 93.38687 | 77.77401 | 14.05681 | 54.31703 | 17.32591 | 16.78431 | 14.29189 | 4.782198 |
| Jarque-Bera | 203454.6 | 141018.8 | 3706.320 | 66557.17 | 6243.125 | 5382.580 | 4115.652 | 133.6397 |
| Probability | 0.000000 | 0.000000 | 0.000000 | 0.000000 | 0.000000 | 0.000000 | 0.000000 | 0.000000 |
| Sum | 24822.83 | 32066.70 | 78.51690 | 12.84793 | 26221.91 | 89037.02 | 164.0861 | 1724.990 |
| Sum Sq. Dev. | 4219097. | 8439034. | 14.35357 | 0.983534 | 6217899. | 24755592 | 135.3906 | 8541.184 |
| Observations | 580 | 580 | 580 | 580 | 580 | 580 | 580 | 580 |

Source: Prepared by the author based on EVIEWS 9 outputs.

From the following table, we can conclude the following:

- The minimum value of the PAR over all the sample is 0.23 while the maximum is 1233, with average = 42.8 and standard deviation = 85.4.
- The minimum value of the PAN over all the sample is 0.91 while the maximum is 1524, with average = 55.3 and standard deviation = 121.7
- The minimum value of the RDE over all the samples is 0 while the maximum is 0.534, with average = 0.022 and standard deviation = 0.041.
- The minimum value of the PRD over all the samples is 0.28 while the maximum is 660.3, with average = 45.2 and standard deviation = 103.6.
- The minimum value of the THE overall the sample is 0.0016 while the maximum is 2.9, with average = 0.28 and standard deviation = 0.48.
- The minimum value of the STJ over all the sample is 0.69 while the maximum is 1966, with average = 153.5 and standard deviation = 207.8.
- The minimum value of the EDU over all the sample is 0 while the maximum is 1.27, with average = 0.135 and standard deviation = 0.157.
- The minimum value of the GDP per capita growth overall the samples is −14.351 while the maximum is 13.636, with average = 2.974 and standard deviation = 3.841. The following figures show the highest and lowest averages of innovation variables for all developing countries during the period 1990–2018, shown in Figures 3–10. We can conclude the following: Mexico has the highest average PAR, followed by

Hungary, while Tunisia has the lowest one (Figure 3). The lowest. Figure 4 shows that Argentina had the highest average for PAN. Regarding Figure 5, Hungary has the highest average RDE, while China and Indonesia have the lowest averages. Figure 6 shows that Hungary has the highest PRD average, while China, Indonesia, and the Philippines have the lowest averages. According to Figures 7 and 8, Hungary has the highest average for THE and DUE., While China has the lowest average of these two variables. Figure 9 shows that Poland had the highest average for STJ, while the Philippines had the lowest average. China recorded the highest average per capita GDP, while Argentina had the lowest (Figure 10).

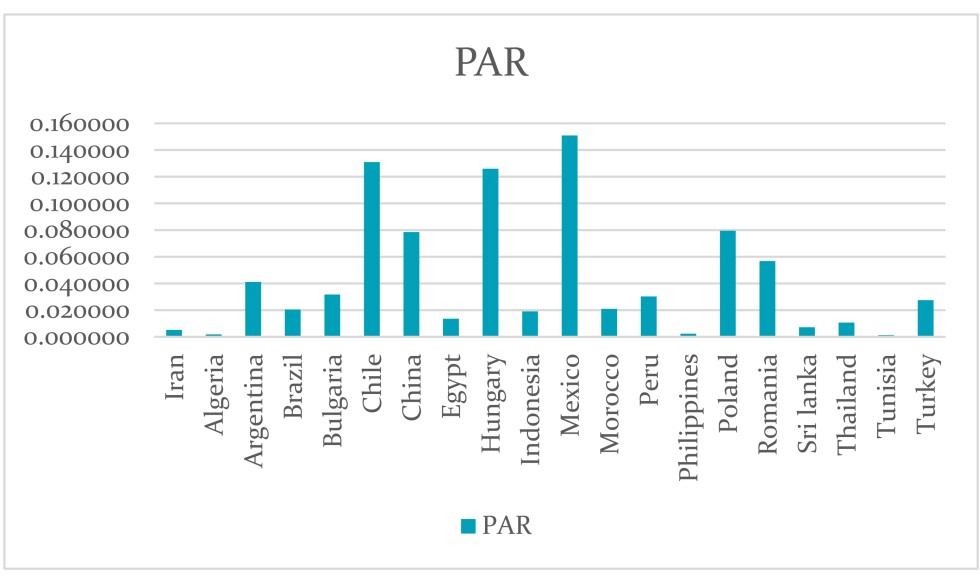

**Figure 3.** Average of PAR across countries.

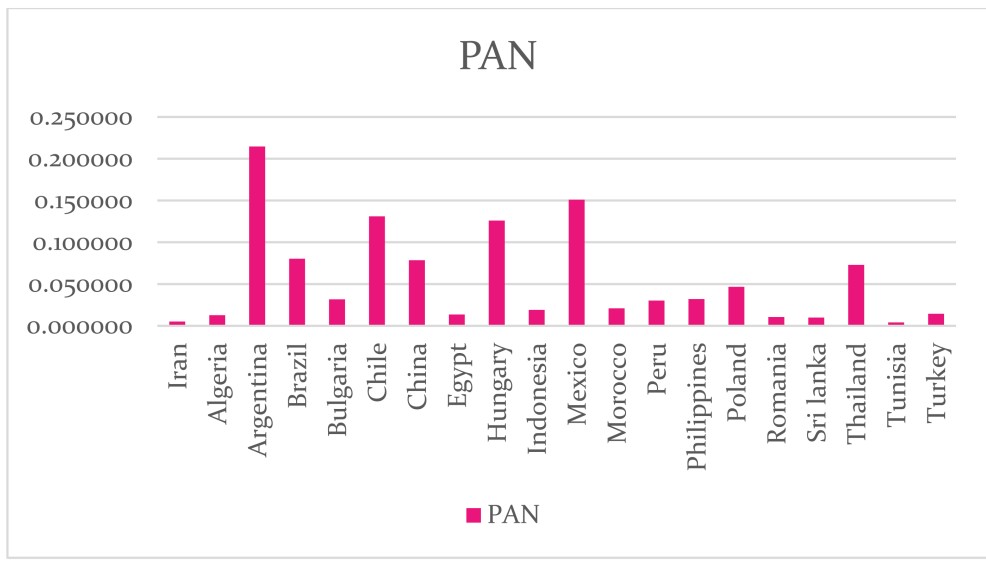

**Figure 4.** Average of PAN across countries.

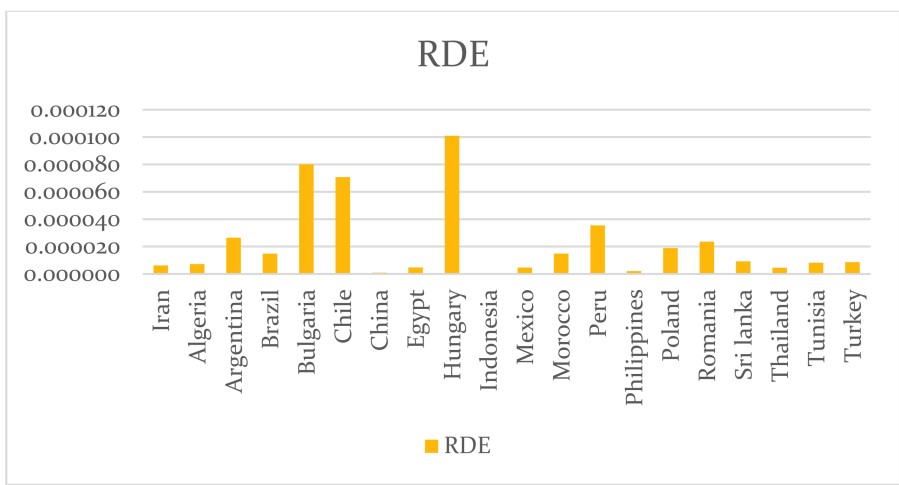

**Figure 5.** Average of RDE across countries.

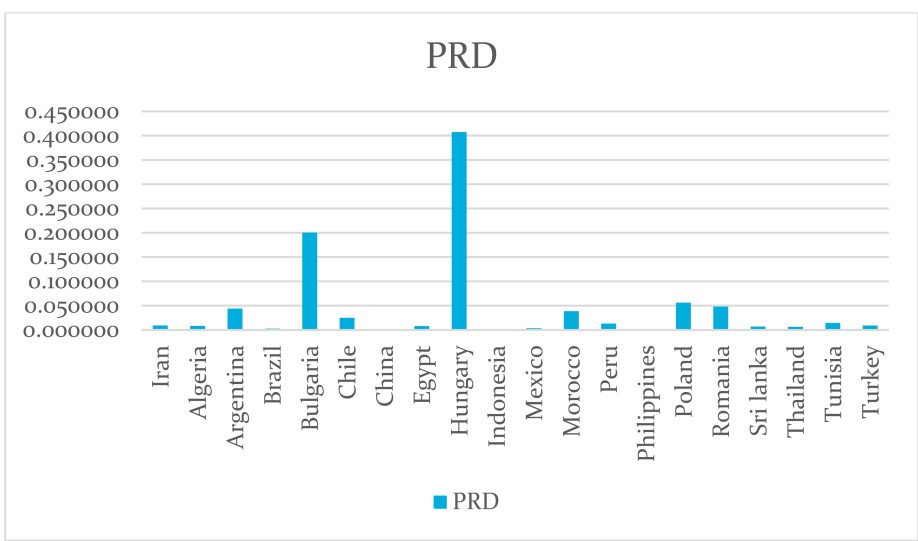

**Figure 6.** Average of PRD across countries.

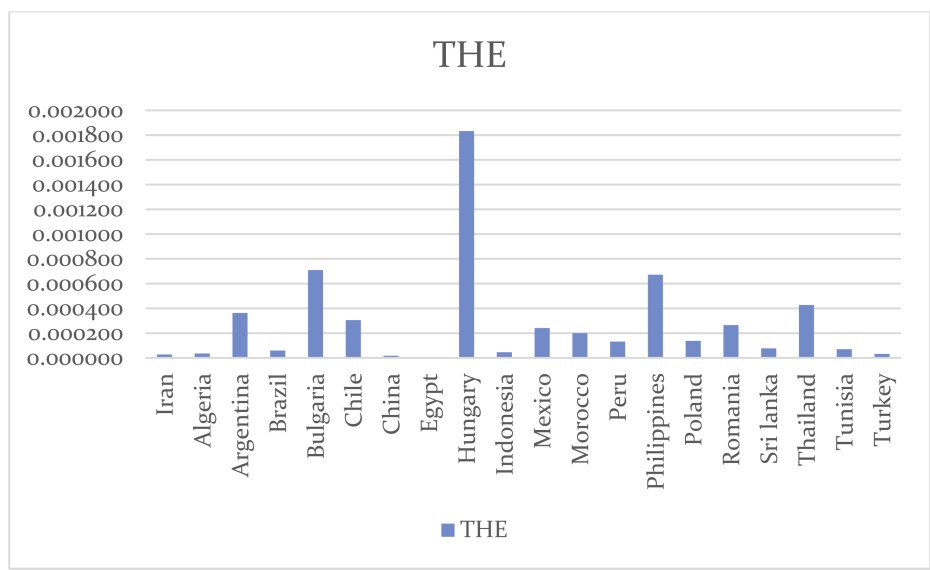

**Figure 7.** Average of THE across countries.

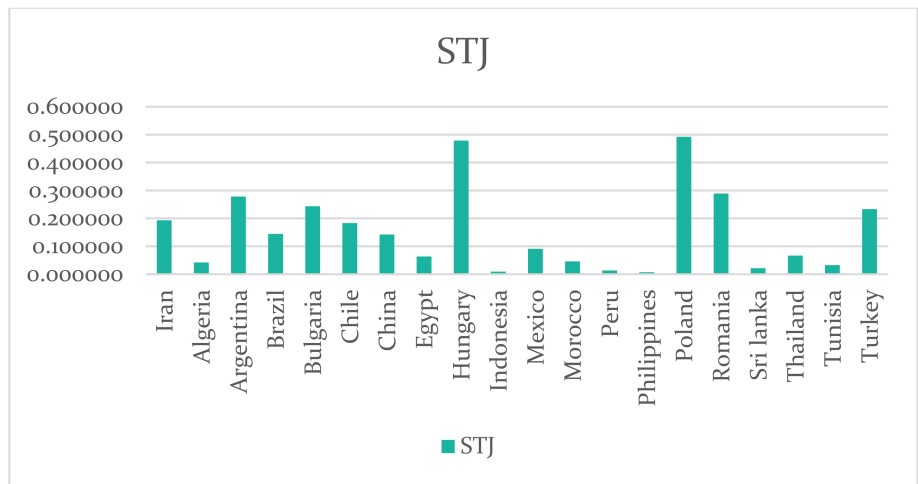

**Figure 8.** Average of STJ across countries.

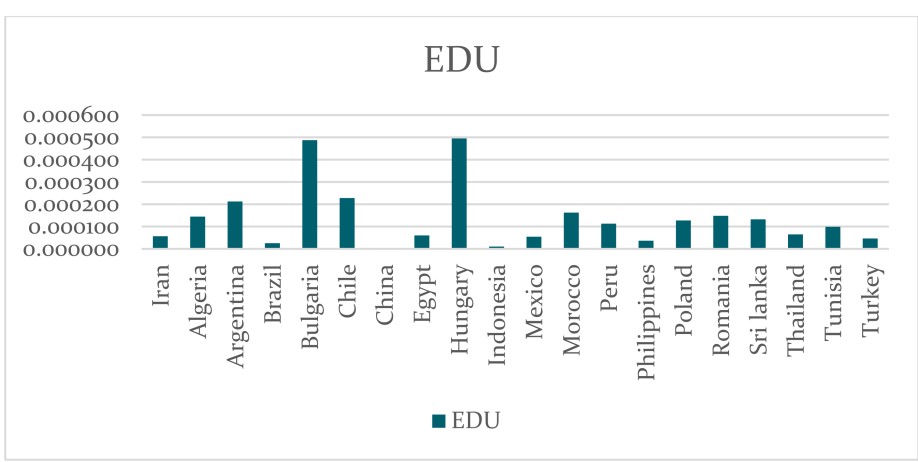

**Figure 9.** Average of EDU across countries.

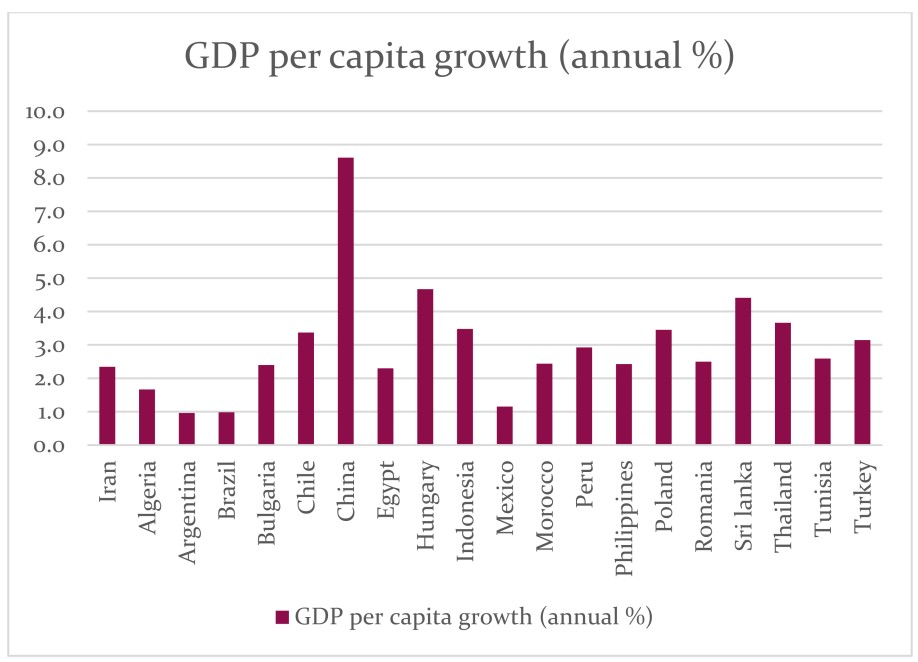

**Figure 10.** Average of GDP per capita across countries.

Table 2 reveals the explanatory variables, including number of patents for residents, non-resident, and educational spending, exhibiting a negative association with GDP. Moreover, the negative correlations imply that the level of GDP reduces as they increase, and vice versa. The positive correlations are shown between the number of researchers in R&D, high-tech exports, and scientific and technical research papers. This simply means that GDP increases as they increase, and vice versa. The results indicate that economic growth and technological innovation are in a positive relationship. The unit root test (UR) is used to determine the stability of the time-series data for the variables included in the model and at the level of differences this stability is achieved, and through that, the integration order is determined for the model variables. Tables 3 and 4 show the summary results of the unit root test, for the variables in their original form or after making its first difference, through the augmented Dickey-Fuller test.

**Table 2.** Correlation matrix.

| Items | GDP_PER_CAPITA_GROWTH__A | PAR | PAN | RDE | PRD | STJ | THE | EDU |
|---|---|---|---|---|---|---|---|---|
| GDP_PER_CAPITA_GROWT | 1.000000 | −0.004692 | −0.047889 | 0.011245 | 0.061354 | 0.038487 | 0.056533 | −0.009516 |
| PAR | −0.004692 | 1.000000 | 0.746965 | 0.204002 | 0.149051 | 0.324532 | 0.311667 | 0.277678 |
| PAN | −0.047889 | 0.746965 | 1.000000 | 0.217452 | 0.142241 | 0.402496 | 0.367943 | 0.441065 |
| RDE | 0.011245 | 0.204002 | 0.217452 | 1.000000 | 0.583682 | 0.389245 | 0.494083 | 0.635408 |
| PRD | 0.061354 | 0.149051 | 0.142241 | 0.583682 | 1.000000 | 0.539553 | 0.797042 | 0.739735 |
| STJ | 0.038487 | 0.324532 | 0.402496 | 0.389245 | 0.539553 | 1.000000 | 0.501469 | 0.552201 |
| THE | 0.056533 | 0.311667 | 0.367943 | 0.494083 | 0.797042 | 0.501469 | 1.000000 | 0.685608 |
| EDU | −0.009516 | 0.277678 | 0.441065 | 0.635408 | 0.739735 | 0.552201 | 0.685608 | 1.000000 |

Source. Authors' computations using E-View Version 9.0.

As an initial step of the analysis before the causality, tests validate the stationarity assumption. The stationarity assumption was tested using the augmented Dickey–Fuller (ADF) test. The ADF test is one of the cited unit root tests in the literature and is commonly used. The ADF test was applied to determine whether the data series was stationary (no unit root) or not, by calculating the respective statistics and $p$-values in the main level, and this is conducted for each country.

Table 3 displays the results of the ADF test using the unit root ADF test at each of the individual countries and using the statistical program (EViews). The results of the stability tests in the plane (in a model with a single constant and direction, with a single constant, without a single constant and direction) indicate that all-time series is unstable in the plane where the corresponding probability of these tests in most of the models was greater than the significance limit (0.05) or (0.1). As for stability tests in the first differences, the results indicate that the remaining time series are all stable in the first differences in all models, that is, it is (1) l, where the corresponding probability of these tests was less than the significance limit (0.05 or 0.1). The stability of time series at the level and in the first differences means there is the possibility of a co-integration relationship between these time series, as shown in Table 4.

**Table 3.** Results of the augmented Dickey–Fuller (ADF) test for unit root variable.

| Country | PARLV (FD) | PANLV (FD) | RDELV (FD) | PRDLV (FD) | THELV (FD) | STJLV (FD) | EDULV (FD) | GDP per Capita Growth (Annual %) LV (FD) |
|---|---|---|---|---|---|---|---|---|
| Algeria | 2.158 (20.7 ***) | 2.206 (24.03 ***) | 22.12 *** | 0.217 (34.836) | 8.37 ** | $5.7 \times 10^{-5}$ 15.7 *** | 12.17 *** | 1.798 (17.034 ***) |
| Argentina | 18.42 *** | 18.42 *** | 20.12 *** | 18.79 *** | 19.445 *** | 18.42 *** | 18.46 *** | 0.289 (14.99 ***) |
| Brazil | 6.373 ** | 1.127 (13.05 ***) | 1.17 (23.2 ***) | 1.287 (31.22 ***) | 5.298 * (8.62 ***) | 0.00784 (16.87 ***) | 5.64 * (26.29 ***) | 0.573 (8.63 **) |
| Bulgaria | 0.911 (17.6 ***) | 0.911 (17.6 ***) | 2.503 (22.9 ***) | 0.031 (14.86 ***) | 0.0237 (15.06 ***) | 0.02188 (20.067 ***) | 1.535 (24.89 ***) | 0.99145 (7.67 **) |
| Chile | 7.498 ** | 7.498 ** | 0.212 (15.9 ****) | 1.59 (34.1 ***) | 1.813 (25.85 ***) | 0.0025 (18.42 ***) | 1.36 (7.22 ***) | 0.7916 (6.304 **) |
| China | (13.6 ***) | 13.63 *** | 0.036 (19.8 ***) | 0.319 (16.9 ***) | 1.9007 (6.3 ***) | 0.0011 (14.085 ***) | 7.43 ** | 0.3077 (15.12 ***) |
| Egypt | 2.52 (16.96 ***) | 5.52 (16.97 ***) | 9.553 *** | 3.96 (18.95 ***) | 12.8 *** | 0.01555 (33.65 ***) | 4.826 * (18.42 ***) | 2.2902 (21.65 ***) |
| Hungary | 0.9131 (8.27 **) | 2.37 (8.27 **) | 2.5 (14.47 ***) | 0.227 (19.47 ***) | 2.05 (13.25 ***) | 1.83 (23.47 ***) | 1.79 (9.62 ***) | 0.02018 (19.29 ***) |
| Indonesia | 3.011 (23.8 ***) | 3.011 (26.5 ***) | 1.39 (32.5 ***) | 2.9 (29.86 ***) | 2.33 (18.86 ***) | 0.0039 (24.82 ***) | 3.0074 (25.9 ***) | 0.40005 (18.42 ***) |
| Iran | 0.544 (19.3 ***) | 0.544 (19.3 ***) | 17.08 *** | 4.97 * (17.47 ***) | 2.24 (21.17 ***) | 0.01927 (6.911 **) | 1.039 (20.83 ***) | 4.568 (11.74 ***) |
| Mexico | 16.57 *** | 16.57 *** | 1.79 (22.4 ***) | 17.025 *** | 16.0697 *** | 15.74 *** | 16.56 *** | 2.839 (15.006 ***) |
| Morocco | 0.00061 (6.66**) | 0.00061 (6.66 **) | 18.4 *** | 0.00053 (15.73 ***) | 1.56 (11.16 ***) | 2.0E-05 (31.9 ***) | 9.016 ** | 0.233 (11.956 ***) |
| Peru | 4.08 (15.9 ***) | 5.212 * (16.18 ***) | 8.1 ** | 3.34 (13.67 ***) | 5.11 * (18.74 ***) | 3.1E09 (34.064 ***) | 5.234 * (30.51 ***) | 0.17077 (9.222 ***) |
| Philippines | 0.13564 (8.71 ***) | 9.708 ** | 0.143 8.4 ** | 1.37 (17.6 ***) | 1.28 (9.38 ***) | 5.4E-08 (24.77 ***) | 2.799 (15.55 ***) | 0.1331 (13.0118 ***) |
| Poland | 0.5878 (12.94 ***) | 0.859 (12.78 ***) | 0.143 (8.4 ***) | 13.22 *** | 0.0338 (10.33 ***) | 0.024 (16.76 ***) | 5.83 * (11.42 ***) | 2.9957 (13.548 ***) |

Table 3. *Cont.*

| Country | PARLV (FD) | PANLV (FD) | RDELV (FD) | PRDLV (FD) | THELV (FD) | STJLV (FD) | EDULV (FD) | GDP per Capita Growth (Annual %) LV (FD) |
|---|---|---|---|---|---|---|---|---|
| Romania | 14.93 *** | 0.45044 (22.47 ***) | 5.62 * (27.7 ***) | 40.25 *** | 1.288 (10.96 ***) | 0.44 (12.8 ***) | 5.38 * (31.14 ***) | 0.10418 (11.128 ***) |
| Sri Lanka | 0.816 (33.47 ***) | 2.433 (20.9 ***) | 3.22 (13.8 ***) | 0.618 (13.98 ***) | 4.37 (10.6 ***) | 6.2E-07 (7.06 **) | 1.88 (22.42 ***) | 0.03594 (6.326 **) |
| Thailand | 2.05 (26.26 ***) | 6.65 ** | 1.46 (6.003 **) | 3.5E-06 (18.97 ***) | 6.11573** | 0.0002 (13.32 ***) | 11.55 ** | 0.2262 (7.845 ***) |
| Tunisia | 0.823 (22.3 ***) | 0.76 (8.196 ***) | 1.066 (9.85 ***) | 0.13077 (11.196 ***) | 0.583 (23.55 ***) | 0.00112 (28.24 ***) | 3.68 (27.87 ***) | 2.050 (11.174 ***) |
| Turkey | 0.0064 (12.72**) | 2.841 (7.123 **) | 1.137 (22.7 ***) | 0.01690 (15.05 ***) | 6.077 ** | 0.313 (30.084 ***) | 2.66 (30.33 ***) | 0.528 (16.75 ***) |

* 10%, ** 5%, *** 1% significance. ADF t-statistic reported. Note: The ADF tests include an intercept. The appropriate lag lengths were selected according to the Schwartz Bayesian criterion (SIC). Source. Authors' computations using E-View Version 9.0.

**Table 4.** The degree of integration of the variables of the model under study.

| Variables | PAR | PAN | RDE | PRD | THE | STJ | EDU | GDP |
|---|---|---|---|---|---|---|---|---|
| Degree of integration | I (1) | I (1) | I (1) | I (1) | I (1) | I (1) | I (1) | I (1) |

Prepared by researchers based on Table 3.

The appropriate lag lengths were selected according to the Schwartz Bayesian criterion (SIC).

It is clear from Table 5 and Equation (8) that the results of the Granger causality test in each country are as follows As shown in Appendix A: (1) Regarding the PAN, PAN was a significant Granger cause of GDP per capita in each of Algeria, Chile, Egypt, Poland, Thailand, and Turkey, with 95% confidence, while in Iran, with 90% confidence. In addition, GDP per capita was a Granger cause of PAN in Brazil, Bulgaria, China, Indonesia, Morocco, Romania, and Tunisia. (2) Regarding the PAR, Par was a significant Granger cause of GDP per capita in each of Chile, Egypt, with 95% confidence, while in Argentina with 90% confidence. Additionally, GDP per capita Granger caused PAR in each of Bulgaria, China, Indonesia, Morocco, Poland, with 95% confidence. (3) Regarding the PRD, PRD was a significant Granger cause of GDP per capita in each of Argentina, Indonesia, and Iran, with 95% confidence. In addition, GDP per capita Granger caused PRD in Brazil, with 95% confidence, and in Argentina with 90% confidence. (4) Regarding the RDE, RDE was a significant Granger-cause of GDP per capita in each of Argentina, Hungary, Iran, Morocco, Philippines, and Tunisia, with95% confidence. GDP per capita Granger-caused RDE in Chile, China, Egypt, Hungary, and Turkey, with 95% confidence, and in Poland and Thailand, with 90% confidence. (5) Regarding the STJ, STG is a significant Granger cause of GDP per capita in each of Argentina, Brazil, Bulgaria, China, Hungary, Romania, Thailand, and Turkey, with 95% confidence, and with 90% confidence in Iran. In addition, GDP per capita Granger caused STJ in Chile, Morocco, Peru, Romania, Thailand, and Tunisia, with 95% confidence, and in Argentina and Hungary, with 90% confidence. (6) Regarding THE, THE significantly Granger caused GDP per capita in each of Algeria, Bulgaria, China, Hungary, Iran, and Thailand, with 95% confidence. GDP per capita Granger-caused THE in Chile, Indonesia, Poland, and Tunisia, with 95% confidence. (7) Regarding EDU, EDU was a significant Granger cause of GDP per capita in each of Brazil, Egypt, Philippines, and Thailand, with 95% confidence, and in Argentina, with 90% confidence. In addition, GDP per capita Granger caused EDU in Chile, Egypt, the Philippines, and Tunisia, with 95% confidence, and with 90% in Brazil. Analysis of data by panel: This type of model known as a "mixed effects model" in the context of panel data. The basic cross-sectional dependent effect may be set, be random or both. It is understood that the estimation process of such a model depends on the main assumption, which is that the $\pi i$ errors are independent and normally distributed with the mean vector 0 and the covariance matrix $\sigma_\varepsilon^2 \sigma$, respectively. Moreover, $bi's$ random effects are independent of $\varepsilon i's$, and usually distributed with mean vector 0 and covariance matrix. Therefore, before running this model we need to verify the normality.

**Table 5.** Results of the Granger causality test for developing countries 1990–2018.

| | Null Hypothesis: | Algeria | Argentina | Brazil | Bulgaria | Chile |
|---|---|---|---|---|---|---|
| PAN | PAN does not Granger Cause GDP_PER_CAPITA | 3.45139 ** | 1.85640 | 0.02088 | 0.55365 | 6.42670 ** |
| | GDP_PER_CAPITA does not Granger Cause PAN | 0.65959 | 1.20757 | 3.50598 ** | 4.24517 ** | 0.16010 |
| PAR | PAR does not Granger Cause GDP_PER_CAPITA | 0.28866 | 2.75374 * | 1.05750 | 0.55365 | 6.42670 ** |
| | GDP_PER_CAPITA does not Granger Cause PAR | 0.54733 | 1.69911 | 0.92123 | 4.24517 ** | 0.16010 |
| PRD | PRD does not Granger Cause GDP_PER_CAPITA | 0.41770 | 3.75307 ** | 0.61420 | 2.08852 | 0.93974 |
| | GDP_PER_CAPITA does not Granger Cause PRD | 1.35729 | 2.52407 * | 3.58595 ** | 1.31153 | 0.70005 |
| RDE | RDE does not Granger Cause GDP_PER_CAPITA | 0.19865 | 3.80148 ** | 0.93278 | 2.25745 | 0.25433 |
| | GDP_PER_CAPITA does not Granger Cause RDE | 0.11666 | 1.31620 | 1.57160 | 1.08045 | 4.69006 ** |
| STJ | STJ does not Granger Cause GDP_PER_CAPITA | 0.88593 | 3.80122 ** | 3.85557 ** | 24.2234 *** | 0.70643 |
| | GDP_PER_CAPITA does not Granger Cause STJ | 0.72180 | 2.78491* | 0.82624 | 0.47384 | 8.69955 *** |
| THE | THE does not Granger Cause GDP_PER_CAPITA | 5.55960 ** | 2.11193 | 0.48506 | 9.65160 *** | 2.18918 |
| | GDP_PER_CAPITA does not Granger Cause THE | 0.87955 | 1.21826 | 0.26147 | 1.68544 | 5.95758 ** |
| EDU | EDU does not Granger Cause GDP_PER_CAPITA | 0.06448 | 2.50999 * | 3.45232 ** | 0.17195 | 0.54776 |
| | GDP_PER_CAPITA does not Granger Cause EDU | 0.08922 | 1.46903 | 3.39050 * | 2.04251 | 4.74801 ** |
| | Null Hypothesis: | China | Egypt | Hungary | Indonesia | Iran |
| PAN | PAN does not Granger Cause GDP_PER_CAPITA | 0.07652 | 7.85133 *** | 0.11664 | 0.60294 | 3.28536 * |
| | GDP_PER_CAPITA does not Granger Cause PAN | 4.13486 ** | 0.05001 | 1.36300 | 5.20243 ** | 0.25676 |
| PAR | PAR does not Granger Cause GDP_PER_CAPITA | 0.07652 | 7.85133 *** | 0.11664 | 0.60294 | 3.28536 * |
| | GDP_PER_CAPITA does not Granger Cause PAR | 4.13486 ** | 0.05001 | 1.36300 | 5.20243 ** | 0.25676 |
| PRD | PRD does not Granger Cause GDP_PER_CAPITA | 1.59115 | 1.01360 | 1.23194 | 4.82837 ** | 8.85677 *** |
| | GDP_PER_CAPITA does not Granger Cause PRD | 1.71315 | 0.50835 | 6.58574 ** | 0.11963 | 0.09699 |
| RDE | RDE does not Granger Cause GDP_PER_CAPITA | 0.63068 | 0.12053 | 4.35234 ** | 0.01557 | 9.04547 *** |
| | GDP_PER_CAPITA does not Granger Cause RDE | 6.14284 *** | 6.87595 ** | 53.1925 *** | 0.51660 | 0.07771 |
| STJ | STJ does not Granger Cause GDP_PER_CAPITA | 4.29613 ** | 0.02424 | 4.47252 ** | 0.02185 | 3.93442 * |
| | GDP_PER_CAPITA does not Granger Cause STJ | 1.72200 | 1.20223 | 2.96970 * | 2.43003 | 0.50925 |
| THE | THE does not Granger Cause GDP_PER_CAPITA | 5.98152 ** | 0.06169 | 3.48594 * | 1.54441 | 4.39407 ** |
| | GDP_PER_CAPITA does not Granger Cause THE | 0.04888 | 1.60651 | 1.65908 | 6.48804 *** | 0.07888 |
| EDU | EDU does not Granger Cause GDP_PER_CAPITA | 0.08946 | 18.0285 *** | 1.07799 | 1.45638 | 2.43955 |
| | GDP_PER_CAPITA does not Granger Cause EDU | 0.13876 | 6.94753 ** | 0.00019 | 0.39204 | 3.25965 * |
| | Null Hypothesis: | Mexico | Morocco | Peru | Philippines | Poland |
| PAN | PAN does not Granger Cause GDP_PER_CAPITA | 0.33389 | 0.89077 | 0.97908 | 1.32599 | 3.94050 ** |
| | GDP_PER_CAPITA does not Granger Cause PAN | 0.11052 | 3.30388 ** | 0.66307 | 0.14147 | 0.95690 |
| PAR | PAR does not Granger Cause GDP_PER_CAPITA | 0.33389 | 0.89077 | 0.97908 | 1.37618 | 0.58816 |
| | GDP_PER_CAPITA does not Granger Cause PAR | 0.11052 | 3.30388 ** | 0.66307 | 1.62187 | 2.94843 * |
| PRD | PRD does not Granger Cause GDP_PER_CAPITA | 0.28754 | 0.45675 | 0.27895 | 0.33509 | 0.26065 |
| | GDP_PER_CAPITA does not Granger Cause PRD | 0.09579 | 1.24709 | 0.45400 | 0.69119 | 0.80394 |
| RDE | RDE does not Granger Cause GDP_PER_CAPITA | 0.31281 | 3.90725 ** | 0.91969 | 4.55742 ** | 1.05090 |
| | GDP_PER_CAPITA does not Granger Cause RDE | 0.18231 | 0.69901 | 0.36466 | 0.67502 | 3.07919 * |
| STJ | STJ does not Granger Cause GDP_PER_CAPITA | 0.26862 | 0.74834 | 0.09426 | 0.15912 | 2.19014 |
| | GDP_PER_CAPITA does not Granger Cause STJ | 1.67510 | 3.48123 ** | 5.11692 ** | 0.41448 | 0.18275 |
| THE | THE does not Granger Cause GDP_PER_CAPITA | 0.29089 | 1.98610 | 0.93801 | 1.78683 | 0.81335 |
| | GDP_PER_CAPITA does not Granger Cause THE | 0.01289 | 1.64873 | 0.91330 | 0.26876 | 4.74172 ** |
| EDU | EDU does not Granger Cause GDP_PER_CAPITA | 0.32060 | 0.59479 | 0.03313 | 3.61820 ** | 0.04317 |
| | GDP_PER_CAPITA does not Granger Cause EDU | 0.02119 | 1.69819 | 0.17291 | 0.98913 | 1.17654 |
| | Null Hypothesis: | Romania | Sri Lanka | Thailand | Tunisia | Turkey |
| PAN | PAN does not Granger Cause GDP_PER_CAPITA | 2.56142 | 0.13404 | 4.52198 ** | 0.25761 | 5.64758 ** |
| | GDP_PER_CAPITA does not Granger Cause PAN | 6.73806 *** | 0.34464 | 1.67446 | 7.66646 *** | 1.47880 |
| PAR | PAR does not Granger Cause GDP_PER_CAPITA | 2.40498 | 1.79948 | 1.14622 | 3.5948 * | 0.36393 |
| | GDP_PER_CAPITA does not Granger Cause PAR | 0.14059 | 9.57892 *** | 0.80727 | 6.67297 ** | 2.40591 |

**Table 5.** *Cont.*

|  | Null Hypothesis: | Algeria | Argentina | Brazil | Bulgaria | Chile |
|---|---|---|---|---|---|---|
| PRD | PRD does not Granger Cause GDP_PER_CAPITA | 0.85992 | 1.56537 | 1.54357 | 0.04494 | 0.77162 |
|  | GDP_PER_CAPITA does not Granger Cause PRD | 0.29628 | 1.02108 | 1.21407 | 2.59188 | 1.20477 |
| RDE | RDE does not Granger Cause GDP_PER_CAPITA | 1.02146 | 0.27437 | 0.76778 | 9.86050 *** | 0.29961 |
|  | GDP_PER_CAPITA does not Granger Cause RDE | 0.13856 | 1.95295 | 2.69477 * | 1.16887 | 4.26462 ** |
| STJ | STJ does not Granger Cause GDP_PER_CAPITA | 9.11119 *** | 0.98285 | 6.39219 *** | 2.20958 | 8.52400 *** |
|  | GDP_PER_CAPITA does not Granger Cause STJ | 19.9675 *** | 0.50109 | 4.38578 ** | 16.5676 *** | 0.25703 |
| THE | THE does not Granger Cause GDP_PER_CAPITA | 0.35056 | 1.00865 | 8.52687 *** | 0.10476 | 1.62364 |
|  | GDP_PER_CAPITA does not Granger Cause THE | 11.8021 *** | 1.46224 | 0.56740 | 15.6075 *** | 1.12907 |
| EDU | EDU does not Granger Cause GDP_PER_CAPITA | 2.68070 | 0.59774 | 5.85179 *** | 0.00675 | 0.41114 |
|  | GDP_PER_CAPITA does not Granger Cause EDU | 2.94423 | 0.34051 | 0.02474 | 7.53871 ** | 1.06322 |

* 10%, ** 5%, *** 1% significance. F-statistics are reported in the table. Source. Authors' computations using E-View Version 9.0.

The normality test is tested using the Jarque-Bera test for normality, where "data meets normal distribution" and is the null hypothesis for the Jarque-Bera test; the findings are shown in Table 6. For the current analysis, it was found that the presumption of normality was not retained. This is because the test's *p*-value was less than 0.05 with a 95% trust. Therefore, the definition of error in our case does not obey the normal distribution. This is illustrated by the results of testing the normal distribution of the dependent variable using the Jarque-Bera test found in Table 6 and Figure 11.

**Table 6.** Results of the Jarque-Bera test.

| Variable | Statistic | Asymp. Sig. (Two-Tailed) |
|---|---|---|
| GDP per capita growth | 12.326 | 0.0021 |

Source. Authors' computations using E-View Version 9.0.

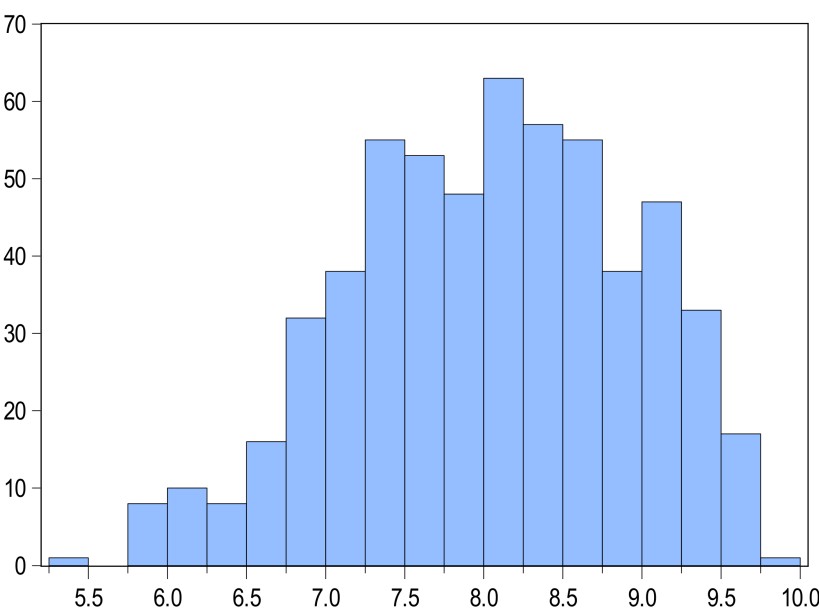

**Figure 11.** The results of the normal distribution test for the dependent variable using the Jarque-Bera test.

From Table 6 and Figure 11, which express the statistical value of the Bera-Jarque test: we find that the probability value (*p*-value) for this statistic, which is equal to (0.021), is less than the level of significance of 5%, and therefore the indicators of innovation during the study period are not subject to a normal distribution, This means that the null hypothesis is rejected and the alternative hypothesis is accepted, and thus the random walk hypothesis is not fulfilled.

Note, Sig (two-tailed) this is the two-tailed *p*-value evaluating the null against an alternative that the mean is not equal to 50. It is equal to the probability of observing a greater absolute value of t under the null hypothesis. If the *p*-value is less than the pre-specified alpha level.

So, the generalized method of moments (GMM) developed by Arellano and Bond (1991) is a good candidate estimation process. Fagerberg et al., and Habibi and Karimi formalized the GMM estimation, which has since become one of the most used estimation methods for economics and finance models. Unlike the estimate of maximum probability (MLE), GMM does not need full knowledge of the data distribution. GMM estimation involves only specified moments deriving from an underlying model. In certain instances, where the data distribution is known, MLE can be computationally burdensome, while GMM can be computationally simple. GMM estimation provides a straightforward means of testing the specification of the proposed model in models for which there are more momentary conditions than model parameters. This is an important, unique feature of GMM estimation. It is well known that when estimating the equations, the GMM method uses the difference of the variables, which are the unconfined compressive strength.

The GMM method has been widely used in recent experimental studies, especially in macroeconomic and finance studies due to its advantages and the capabilities of panel models in using GMM, while it is good at exploiting the variance of time-series data and calculating the individual unobserved effects, thus providing better control of the specificity of all explanatory variables. This contributes to obtaining consistent and unbiased estimations [86–88]. This is illustrated by the results shown in Table 7. The model validity test is a method that uses two tests:

- Sargan-Hansen test or Sargan's test is a statistical test used in the statistical model to assess over-identifying limitations. In other words, it checks whether the instrument variables used are correct. This test's null hypothesis is, "no over-identification." If the null hypothesis is not dismissed, the model is correct.
- The Arellano-Bond method tests whether the errors are correlated. This test's null hypothesis is "no self-correlation". If the null hypothesis is not dismissed, the model is correct. The findings are provided in Tables 7 and 8 and it is possible to infer that there is a significant negative impact of education (percent of expenditure in GDP) on GDP per capita growth, and this effect = −3.5, with 95% confidence, as the *p*-value of the coefficient is less than 5%. Moreover, findings show a significant positive impact of research development and expenditure on GDP per capita growth, with an effect = 0.00269, and with a 95% confidence as the *p*-value of the coefficient is less than 5%. While there is a significant positive impact of scientific and technical journal articles on GDP per capita growth, and this effect = 0.004503, with 95% confidence as the *p*-value of the coefficient is less than 5%. However, there is a significant positive impact of high technology exports on GDP per capita growth, and this effect = 0.740, with 95% confidence as the *p*-value of the coefficient is less than 5%. Finally, there is an insignificant impact of each of PAN, PAR, and RDE on GDP per capita, with 95% confidence as the *p*-value for these coefficients are greater than 5%. Regarding the goodness of fit of the model and analysis of the results of the dynamic model estimation for GMM, the empirical results from estimating the dynamic models of panel data by GMM are good if the estimated values of the regression coefficients of these models by this method are consistent and consistency is achieved with the actual values of the regression coefficients. This can be illustrated by the graph in Figure 12 showing the consistency of the actual, estimated, and residual values with each other.

Additionally, to determine the validity of these variables, Sargan's statistical test was used, and it was greater than 5%. Therefore, the null hypothesis was accepted, which states the quality and suitability of the tools used in the model and the validity of the moment conditions used in the estimation. It was also clear through Sargan's test that the delay variables were valid and that the first-degree differences were statistically acceptable. On the other hand, the statistical value of the Arellano-Bond test for second-order serial correlation between the estimated errors with the first step indicates that the null hypothesis of this test is not rejected, which is the absence of this correlation. This means that the original error term is not sequentially related. This is because the estimation results in Table 7 show that the probability of this test statistic is greater than 5%, equal to (0.9907) that is, accepting the null hypothesis that there is no second-order serial correlation to the random error, and this indicates the validity of the moment constraints used in the estimation.

**Table 7.** Results of estimating the impact of technological innovation on economic growth using the panel generalized method of moments (GMM).

| Variable | Coefficient | Standard Error | t-Statistic | Probability |
|---|---|---|---|---|
| EDU | −3.500127 | 0.390585 | −8.961248 | 0.0000 |
| PAN | −0.000345 | 0.001832 | −0.188131 | 0.8508 |
| PAR | 0.000147 | 0.001754 | 0.084037 | 0.9331 |
| PRD | 0.002690 | 0.001332 | 2.019151 | 0.0440 |
| RDE | −0.686891 | 1.316314 | −0.521829 | 0.6020 |
| STJ | 0.004503 | 0.001514 | 2.974776 | 0.0031 |
| THE | 0.740715 | 0.245014 | 3.023156 | 0.0026 |
| Effects Specification | | | | |
| Cross-section fixed (first differences) | | | | |
| Mean dependent var | 0.047455 | S.D. dependent var | | 0.206800 |
| S.E. of regression | 0.501957 | Sum squared resid | | 139.3344 |
| J-statistic | 13.90493 | Instrument rank | | 21 |
| Prob (J-statistic) | 0.456818 | | | |

Arellano-Bond Serial Correlation Test. Equation: EQ01. Sample: 1990 2018. Included observations: 560.

**Table 8.** Arellano-Bond Serial Correlation Test.

| Test order | m-Statistic | rho | SE (rho) | Prob. |
|---|---|---|---|---|
| AR (2) | −0.011694 | −0.320535 | 27.409643 | 0.9907 |

Source: Prepared by researchers based on the results of EViews 9.

The results of the estimation in the above table showed the statistical significance of four variables (EDU, PRD, STJ, and THE), where the probability value was less than 5%. The estimation results showed the following:

There is a negative and significant effect of 1% for education (EDU): the percent of education expenditure in GDP. This means that an increase in spending on education by 1% leads to a decrease in economic growth by 3.5%, this does not fit with various theoretical analyses that have considered education spending as a driver of economic growth. This is due to developing countries still requiring more spending on education infrastructure so that innovation brings its expected results. There is a positive and significant effect of research and development expenditure (PRD) on economic growth, since every 1% increase in research and development expenditure results in an increase in GDP per capita by 0.27%. Having a positive and significant impact on economic growth, each 1% increase in scientific and technical journal articles (STJ) increases GDP per capita by 0.45%. As for the variable of high technology exports (THE), which has a positive and noticeable impact, an increase in this percentage by 1% leads to an increase in per capita GDP by 0.74%. This means that the variables of technological innovation (researchers' development and expenditure, scientific

and technical journals, and high technology exports) had an important role in influencing the economic growth in the countries under study, and this is consistent with the various theoretical analyses that considered these variables as a driving factor for economic growth. On the other hand, as for the variables (patent fields by non-residents (PAN), patent fields by residents (PAR), and researcher scholars in research and development activities (RDE)), all estimation attempts were unsuccessful in finding an important and significant relationship showing the extent to which the impact of technological innovation contributes to economic growth. Where it turned out that there is a weak and non-significant negative relationship for the variable, patent fields by non-residents (PAN), which means that the increase by 1% in the percentage (PAN). This is accompanied by a decrease in economic growth by 0.00034%. As for the variable, patent fields by residents, its effect was positive and insignificant, and its contribution was very weak, as the increase in this percentage would not lead to high economic growth. As for the variable of researchers in research and development activities, its relationship to economic growth was negative, strong, and not significant, meaning that an increase of 1% in the proportion of researchers in research and development is accompanied by a decrease in economic growth of 0.69%, but here there is no effect because it is not significant in the model.

From Figure 12, we notice a match between the current and estimated values, which indicates the existence of stability in the model.

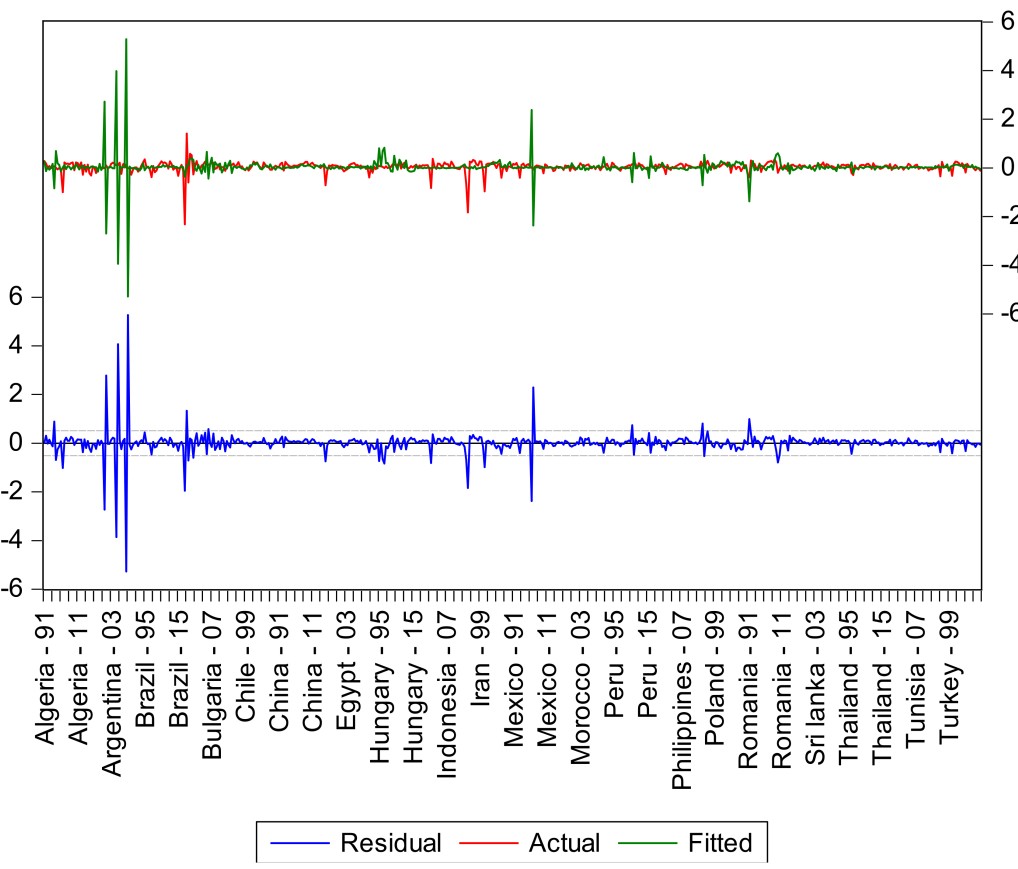

**Figure 12.** Graphic representation of current and estimated values for study variables. Source: Prepared by researchers based on outputs EViews 9.

## 5. Discussion

This study aimed to measure the causality between technological innovation (expenditure on education (EDU); patent fields by residents (PAR); patent fields by non-residents (PAN); researcher scholars in research and development activities (RDE); researchers' development and expenditure (PRD); high technology exports (HTE); and scientific and technical

journals (STG) as independent variables on economic growth in 20 developing countries for the period 1990–2018. Using the economic measurement of the panel data, the study concluded that the economic growth of the countries under study is linked to a causality relationship with some technological innovation indicators. Through the above, we found that Mexico had the highest average PAR, followed by Hungary, while Tunisia had the lowest average. Hungary had the highest average RDE, while China and Indonesia have the lowest averages. Hungary had the highest average PRD, while China, Indonesia, and the Philippines had the lowest averages. Hungary had the highest average for THE and DUE, while China has the lowest average of these two variables. Poland had the highest STJ average, while the Philippines had the lowest. China recorded the highest average per capita GDP, while Argentina recorded the lowest.

The results also indicated for the stability tests in the plane (in a model with a single constant and direction, with a single constant, and without a single constant and direction) that the all-time series is unstable in the plane where the corresponding probability of these tests in most of the models was greater than the significance limit (0.05) or (0.1). As for stability tests in the first differences, the results indicate that the remaining time series are all stable in the first differences in all models, that is, it is (1) l, where the corresponding probability of these tests was less than the significance limit (0.05 or 0.1). The stability of the time series at the level and in the first differences means there is the possibility of a co-integration relationship between these time series.

The results of the estimation showed the statistical significance of four variables (EDU, PRD, STJ, and THE). The estimation results showed there to be a negative and significant effect of expenditure education (EDU). This means that an increase in spending on education leads to a decrease in economic growth, this does not fit with various theoretical analyses that have considered education spending as a driver of economic growth. This is due to developing countries still needing to spend more on education infrastructure so that innovation brings its expected results. There was a positive and significant effect of research and development expenditure (PRD) on economic growth. In addition, scientific and technical journal articles (STJ) showed a positive and significant impact on economic growth. The variable of high technology exports (THE) had a positive and noticeable impact. This means that the variables of technological innovation (researchers' development and expenditure, scientific and technical journals, and high technology exports) had an important role in influencing the economic growth in the countries under study, and this is consistent with the various theoretical analyses that considered these variables as a driving factor for economic growth.

In addition, we found that as for the variables (patent fields by non-residents (PAN), patent fields by residents (PAR), and researcher scholars in research and development activities (RDE)), all estimation attempts were unsuccessful in finding an important and significant relationship showing the extent to which the impact of technological innovation contributes to economic growth. Where it turned out that there is a weak and non-significant negative relationship for the variable patent fields by non-residents (PAN). As for the variable patent fields by residents, its effect was positive and insignificant, and its contribution was very weak, as the increase in this percentage would not lead to high economic growth. As for the variable of researchers in research and development activities, its relationship to economic growth was negative, strong, and not significant. Nevertheless, here there is no effect because it is not significant in the model. Therefore, the results were consistent with what was reported in previous (Abdelaoui. et al. [57], Lomachynska and Podgorna, [58], Pece et al. [59], Solomon et al. [60] and Tuna et al. [56]).

## 6. Conclusions and Recommendations

In this paper, we examined the relationship between technological innovation and economic growth over the period 1990–2018. The study was conducted in 20 developing countries. Comparison was carried out between countries, considering the growth rate of each country's per capita GDP (the growth of the country's economy is measured as a

percentage increase in the gross national product per capita), and the independent variables were education expenditures and patents by residents and non-residents, research and development expenditures, and researchers in research and development activities high-tech, export, and scientific and technical journal articles. The study concluded that the variables relied upon have a positive impact on economic growth. The results showed the significant positive impact of research and spending development, scientific and technical journals, and high-tech exports on per capita GDP growth, while PAN, PAR, and RDE had little effect on GDP per capita; however, the data show Mexico to have the highest average PAR, while Tunisia had the lowest. Argentina had the highest average PAN, while Tunisia had the lowest. Hungary had the highest average RDE, while China and Indonesia had the lowest averages. Hungary had the highest PRD average while the lowest averages were found in China, Indonesia, and the Philippines. As for THE, and DUE, Hungary had the highest average. The lowest value of these two variables was in China. Poland had the highest STJ score while the lowest was in the Philippines. China had the largest average per capita GDP, while the lowest was found for Argentina. In short, we conclude that a country with a high rate of investment in R&D, high-tech exports, patent rights, and science and technology journal articles will be highly developed, and that a country's economy will be very innovative. Hence, the level and structure of innovation should not be overlooked as it plays a fundamental role in stimulating economic growth.

Despite this contribution, as with any research, it comes with some limitations. First, given the method of data collection chosen, most developing countries imply a lack of data and a fallacy in that data. This might be considered a limitation of the study. Secondly, most developing countries are still importing technology from developed countries that may not suit their environment, and this leads to not reaping the fruits of technological innovation expected to be obtained. Finally, our study found that the economic problem today is based on the abundance of information and not the traditional scarce resources because of technological innovation. Moreover, economic growth has become the decisive element in all aspects of economic activity, and knowledge has become the basis for any economic or social growth, consequently, the world has shifted from research and collision in order to source scarce resources to search for and control as many knowledge sources as possible.

Therefore, future research should look at how to direct economic resources towards knowledge industries in a manner equivalent to the volume of resources directed towards investments in the sectors of construction, tourism, sports, and entertainment. Furthermore, there is a need to search for ways to support scientific research and researchers in the field of knowledge technologies and increase the volume of spending on scientific research so that it constitutes a good percentage of Gross National Product, which has a positive impact on the country's national economy. Based on previous analyses and research conclusions, the following suggestions have been articulated to provide the basis for improving levels of technological innovation in developing countries. Based on the results of the study, the following recommendations can be made: (1) for the policymakers, strategies should be adopted to improve high-tech export rather than exporting raw and primary goods; (2) the government should provide appropriate funding for R&D performed in the public sector, especially in the higher education sector; (3) the policy of economic openness for developing countries should be encouraged by taking reform measures at various levels in order to benefit from trade openness to the outside world in the field of innovation, and thus support economic growth; and (4) work should be carried out to create an environment conducive to innovation in developing countries, by expanding spending on research and development, as well as by protecting intellectual property rights.

**Author Contributions:** Methodology, M.M.A.M.; formal analysis, M.M.A.M.; writing—original draft preparation, G.N.; writing—review and editing, P.L. All authors have read and agreed to the published version of the manuscript.

**Funding:** This research received no external funding.

**Institutional Review Board Statement:** Not applicable.

**Informed Consent Statement:** Not applicable.

**Data Availability Statement:** Publicly available datasets were analyzed in this study. These data can be found here: https://databank.worldbank.org/source/world-development-indicators (accessed on 15 January 2022).

**Conflicts of Interest:** The authors declare no conflict of interest.

## Appendix A

**Table A1.** The results of the Granger causality test in each country.

| Pairwise Granger Causality Tests. Sample: 1990 2018 IF COUNTRY1 = 1 | | | |
|---|---|---|---|
| **Null Hypothesis:** | **Obs** | **F-Statistic** | **Prob.** |
| PAN does not Granger Cause GDP_PER_CAPITA | 27 | 3.45139 | 0.0497 |
| GDP_PER_CAPITA does not Granger Cause PAN | | 0.65959 | 0.5270 |
| PAR does not Granger Cause GDP_PER_CAPITA | 27 | 0.28866 | 0.7521 |
| GDP_PER_CAPITA does not Granger Cause PAR | | 0.54733 | 0.5862 |
| PRD does not Granger Cause GDP_PER_CAPITA | 27 | 0.41770 | 0.6637 |
| GDP_PER_CAPITA does not Granger Cause PRD | | 1.35729 | 0.2781 |
| RDE does not Granger Cause GDP_PER_CAPITA | 27 | 0.19865 | 0.8213 |
| GDP_PER_CAPITA does not Granger Cause RDE | | 0.11666 | 0.8904 |
| STJ does not Granger Cause GDP_PER_CAPITA | 27 | 0.88593 | 0.4265 |
| GDP_PER_CAPITA does not Granger Cause STJ | | 0.72180 | 0.4970 |
| THE does not Granger Cause GDP_PER_CAPITA | 27 | 5.55960 | 0.0111 |
| GDP_PER_CAPITA does not Granger Cause THE | | 0.87955 | 0.4291 |
| EDU does not Granger Cause GDP_PER_CAPITA | 27 | 0.06448 | 0.9377 |
| GDP_PER_CAPITA does not Granger Cause EDU | | 0.08922 | 0.9150 |
| Pairwise Granger Causality Tests Sample: 1990 2018 IF COUNTRY1 = 2 | | | |
| Null Hypothesis: | Obs | F-Statistic | Prob. |
| PAN does not Granger Cause GDP_PER_CAPITA | 27 | 1.85640 | 0.1713 |
| GDP_PER_CAPITA does not Granger Cause PAN | | 1.20757 | 0.3340 |
| PAR does not Granger Cause GDP_PER_CAPITA | 27 | 2.75374 | 0.0709 |
| GDP_PER_CAPITA does not Granger Cause PAR | | 1.69911 | 0.2010 |
| PRD does not Granger Cause GDP_PER_CAPITA | 27 | 3.75307 | 0.0285 |
| GDP_PER_CAPITA does not Granger Cause PRD | | 2.52407 | 0.0884 |
| RDE does not Granger Cause GDP_PER_CAPITA | 27 | 3.80148 | 0.0273 |
| GDP_PER_CAPITA does not Granger Cause RDE | | 1.31620 | 0.2983 |
| STJ does not Granger Cause GDP_PER_CAPITA | 27 | 3.80122 | 0.0273 |
| GDP_PER_CAPITA does not Granger Cause STJ | | 2.78491 | 0.0689 |
| THE does not Granger Cause GDP_PER_CAPITA | 27 | 2.11193 | 0.1325 |
| GDP_PER_CAPITA does not Granger Cause THE | | 1.21826 | 0.3303 |
| EDU does not Granger Cause GDP_PER_CAPITA | 27 | 2.50999 | 0.0896 |
| GDP_PER_CAPITA does not Granger Cause EDU | | 1.46903 | 0.2547 |
| Pairwise Granger Causality Tests Sample: 1990 2018 IF COUNTRY1 = 3 | | | |
| Null Hypothesis: | Obs | F-Statistic | Prob. |
| PAN does not Granger Cause GDP_PER_CAPITA | 27 | 0.02088 | 0.9794 |
| GDP_PER_CAPITA does not Granger Cause PAN | | 3.50598 | 0.0477 |
| PAR does not Granger Cause GDP_PER_CAPITA | 27 | 1.05750 | 0.3643 |
| GDP_PER_CAPITA does not Granger Cause PAR | | 0.92123 | 0.4128 |

**Table A1.** *Cont.*

| Null Hypothesis: | Obs | F-Statistic | Prob. |
|---|---|---|---|
| PRD does not Granger Cause GDP_PER_CAPITA | 27 | 0.61420 | 0.5501 |
| GDP_PER_CAPITA does not Granger Cause PRD | | 3.58595 | 0.0449 |
| RDE does not Granger Cause GDP_PER_CAPITA | 27 | 0.93278 | 0.4085 |
| GDP_PER_CAPITA does not Granger Cause RDE | | 1.57160 | 0.2302 |
| STJ does not Granger Cause GDP_PER_CAPITA | 27 | 3.85557 | 0.0367 |
| GDP_PER_CAPITA does not Granger Cause STJ | | 0.82624 | 0.4508 |
| THE does not Granger Cause GDP_PER_CAPITA | 27 | 0.48506 | 0.6221 |
| GDP_PER_CAPITA does not Granger Cause THE | | 0.26147 | 0.7723 |
| EDU does not Granger Cause GDP_PER_CAPITA | 27 | 3.45232 | 0.0497 |
| GDP_PER_CAPITA does not Granger Cause EDU | | 3.39050 | 0.0521 |

Pairwise Granger Causality Tests
Sample: 1990 2018 IF COUNTRY1 = 4

| Null Hypothesis: | Obs | F-Statistic | Prob. |
|---|---|---|---|
| PAN does not Granger Cause GDP_PER_CAPITA | 27 | 0.55365 | 0.4638 |
| GDP_PER_CAPITA does not Granger Cause PAN | | 4.24517 | 0.0499 |
| PAR does not Granger Cause GDP_PER_CAPITA | 27 | 0.55365 | 0.4638 |
| GDP_PER_CAPITA does not Granger Cause PAR | | 4.24517 | 0.0499 |
| PRD does not Granger Cause GDP_PER_CAPITA | 27 | 2.08852 | 0.1608 |
| GDP_PER_CAPITA does not Granger Cause PRD | | 1.31153 | 0.2630 |
| RDE does not Granger Cause GDP_PER_CAPITA | 27 | 2.25745 | 0.1455 |
| GDP_PER_CAPITA does not Granger Cause RDE | | 1.08045 | 0.3085 |
| STJ does not Granger Cause GDP_PER_CAPITA | 27 | 24.2234 | 5.E-05 |
| GDP_PER_CAPITA does not Granger Cause STJ | | 0.47384 | 0.4976 |
| THE does not Granger Cause GDP_PER_CAPITA | 27 | 9.65160 | 0.0047 |
| GDP_PER_CAPITA does not Granger Cause THE | | 1.68544 | 0.2061 |
| EDU does not Granger Cause GDP_PER_CAPITA | 27 | 0.17195 | 0.6819 |
| GDP_PER_CAPITA does not Granger Cause EDU | | 2.04251 | 0.1653 |

Pairwise Granger Causality Tests
Sample: 1990 2018 IF COUNTRY1 = 5

| Null Hypothesis: | Obs | F-Statistic | Prob. |
|---|---|---|---|
| PAN does not Granger Cause GDP_PER_CAPITA | 27 | 6.42670 | 0.0179 |
| GDP_PER_CAPITA does not Granger Cause PAN | | 0.16010 | 0.6925 |
| PAR does not Granger Cause GDP_PER_CAPITA | 27 | 6.42670 | 0.0179 |
| GDP_PER_CAPITA does not Granger Cause PAR | | 0.16010 | 0.6925 |
| PRD does not Granger Cause GDP_PER_CAPITA | 27 | 0.93974 | 0.3416 |
| GDP_PER_CAPITA does not Granger Cause PRD | | 0.70005 | 0.4107 |
| RDE does not Granger Cause GDP_PER_CAPITA | 27 | 0.25433 | 0.6185 |
| GDP_PER_CAPITA does not Granger Cause RDE | | 4.69006 | 0.0401 |
| STJ does not Granger Cause GDP_PER_CAPITA | 27 | 0.70643 | 0.4086 |
| GDP_PER_CAPITA does not Granger Cause STJ | | 8.69955 | 0.0068 |
| THE does not Granger Cause GDP_PER_CAPITA | 27 | 2.18918 | 0.1515 |
| GDP_PER_CAPITA does not Granger Cause THE | | 5.95758 | 0.0221 |
| EDU does not Granger Cause GDP_PER_CAPITA | 27 | 0.54776 | 0.4661 |
| GDP_PER_CAPITA does not Granger Cause EDU | | 4.74801 | 0.0390 |

Pairwise Granger Causality Tests
Sample: 1990 2018 IF COUNTRY1 = 6

| Null Hypothesis: | Obs | F-Statistic | Prob. |
|---|---|---|---|
| PAN does not Granger Cause GDP_PER_CAPITA | 27 | 0.07652 | 0.9266 |
| GDP_PER_CAPITA does not Granger Cause PAN | | 4.13486 | 0.0299 |
| PAR does not Granger Cause GDP_PER_CAPITA | 27 | 0.07652 | 0.9266 |
| GDP_PER_CAPITA does not Granger Cause PAR | | 4.13486 | 0.0299 |

**Table A1.** *Cont.*

| Null Hypothesis: | Obs | F-Statistic | Prob. |
|---|---|---|---|
| PRD does not Granger Cause GDP_PER_CAPITA | 27 | 1.59115 | 0.2263 |
| GDP_PER_CAPITA does not Granger Cause PRD | | 1.71315 | 0.2035 |
| RDE does not Granger Cause GDP_PER_CAPITA | 27 | 0.63068 | 0.5416 |
| GDP_PER_CAPITA does not Granger Cause RDE | | 6.14284 | 0.0076 |
| STJ does not Granger Cause GDP_PER_CAPITA | 27 | 4.29613 | 0.0266 |
| GDP_PER_CAPITA does not Granger Cause STJ | | 1.72200 | 0.2019 |
| THE does not Granger Cause GDP_PER_CAPITA | 27 | 5.98152 | 0.0084 |
| GDP_PER_CAPITA does not Granger Cause THE | | 0.04888 | 0.9524 |
| EDU does not Granger Cause GDP_PER_CAPITA | 27 | 0.08946 | 0.9148 |
| GDP_PER_CAPITA does not Granger Cause EDU | | 0.13876 | 0.8712 |

Pairwise Granger Causality Tests
Sample: 1990 2018 IF COUNTRY1 = 7
Lags: 1

| Null Hypothesis: | Obs | F-Statistic | Prob. |
|---|---|---|---|
| PAN does not Granger Cause GDP_PER_CAPITA | 27 | 7.85133 | 0.0097 |
| GDP_PER_CAPITA does not Granger Cause PAN | | 0.05001 | 0.8249 |
| PAR does not Granger Cause GDP_PER_CAPITA | 27 | 7.85133 | 0.0097 |
| GDP_PER_CAPITA does not Granger Cause PAR | | 0.05001 | 0.8249 |
| PRD does not Granger Cause GDP_PER_CAPITA | 27 | 1.01360 | 0.3237 |
| GDP_PER_CAPITA does not Granger Cause PRD | | 0.50835 | 0.4825 |
| RDE does not Granger Cause GDP_PER_CAPITA | 27 | 0.12053 | 0.7314 |
| GDP_PER_CAPITA does not Granger Cause RDE | | 6.87595 | 0.0147 |
| STJ does not Granger Cause GDP_PER_CAPITA | 27 | 0.02424 | 0.8775 |
| GDP_PER_CAPITA does not Granger Cause STJ | | 1.20223 | 0.2833 |
| THE does not Granger Cause GDP_PER_CAPITA | 27 | 0.06169 | 0.8059 |
| GDP_PER_CAPITA does not Granger Cause THE | | 1.60651 | 0.2167 |
| EDU does not Granger Cause GDP_PER_CAPITA | 27 | 18.0285 | 0.0003 |
| GDP_PER_CAPITA does not Granger Cause EDU | | 6.94753 | 0.0142 |

Pairwise Granger Causality Tests
Sample: 1990 2018 IF COUNTRY1 = 8

| Null Hypothesis: | Obs | F-Statistic | Prob. |
|---|---|---|---|
| PAN does not Granger Cause GDP_PER_CAPITA | 27 | 0.11664 | 0.7356 |
| GDP_PER_CAPITA does not Granger Cause PAN | | 1.36300 | 0.2540 |
| PAR does not Granger Cause GDP_PER_CAPITA | 27 | 0.11664 | 0.7356 |
| GDP_PER_CAPITA does not Granger Cause PAR | | 1.36300 | 0.2540 |
| PRD does not Granger Cause GDP_PER_CAPITA | 27 | 1.23194 | 0.2776 |
| GDP_PER_CAPITA does not Granger Cause PRD | | 6.58574 | 0.0167 |
| RDE does not Granger Cause GDP_PER_CAPITA | 27 | 4.35234 | 0.0473 |
| GDP_PER_CAPITA does not Granger Cause RDE | | 53.1925 | 1.E-07 |
| STJ does not Granger Cause GDP_PER_CAPITA | 27 | 4.47252 | 0.0446 |
| GDP_PER_CAPITA does not Granger Cause STJ | | 2.96970 | 0.0972 |
| THE does not Granger Cause GDP_PER_CAPITA | 27 | 3.48594 | 0.0737 |
| GDP_PER_CAPITA does not Granger Cause THE | | 1.65908 | 0.2095 |
| EDU does not Granger Cause GDP_PER_CAPITA | 27 | 1.07799 | 0.3091 |
| GDP_PER_CAPITA does not Granger Cause EDU | | 0.00019 | 0.9892 |

Pairwise Granger Causality Tests
Sample: 1990 2018 IF COUNTRY1 = 9

| Null Hypothesis: | Obs | F-Statistic | Prob. |
|---|---|---|---|
| PAN does not Granger Cause GDP_PER_CAPITA | 27 | 0.60294 | 0.5560 |
| GDP_PER_CAPITA does not Granger Cause PAN | | 5.20243 | 0.0141 |
| PAR does not Granger Cause GDP_PER_CAPITA | 27 | 0.60294 | 0.5560 |
| GDP_PER_CAPITA does not Granger Cause PAR | | 5.20243 | 0.0141 |

**Table A1.** *Cont.*

| Null Hypothesis: | Obs | F-Statistic | Prob. |
|---|---|---|---|
| PRD does not Granger Cause GDP_PER_CAPITA | 27 | 4.82837 | 0.0183 |
| GDP_PER_CAPITA does not Granger Cause PRD | | 0.11963 | 0.8878 |
| RDE does not Granger Cause GDP_PER_CAPITA | 27 | 0.01557 | 0.9846 |
| GDP_PER_CAPITA does not Granger Cause RDE | | 0.51660 | 0.6036 |
| STJ does not Granger Cause GDP_PER_CAPITA | 27 | 0.02185 | 0.9784 |
| GDP_PER_CAPITA does not Granger Cause STJ | | 2.43003 | 0.1113 |
| THE does not Granger Cause GDP_PER_CAPITA | 27 | 1.54441 | 0.2357 |
| GDP_PER_CAPITA does not Granger Cause THE | | 6.48804 | 0.0061 |
| EDU does not Granger Cause GDP_PER_CAPITA | 27 | 1.45638 | 0.2547 |
| GDP_PER_CAPITA does not Granger Cause EDU | | 0.39204 | 0.6803 |

Pairwise Granger Causality Tests
Sample: 1990 2018 IF COUNTRY1 = 10

| Null Hypothesis: | Obs | F-Statistic | Prob. |
|---|---|---|---|
| PAN does not Granger Cause GDP_PER_CAPITA | 27 | 3.28536 | 0.0819 |
| GDP_PER_CAPITA does not Granger Cause PAN | | 0.25676 | 0.6168 |
| PAR does not Granger Cause GDP_PER_CAPITA | 27 | 3.28536 | 0.0819 |
| GDP_PER_CAPITA does not Granger Cause PAR | | 0.25676 | 0.6168 |
| PRD does not Granger Cause GDP_PER_CAPITA | 27 | 8.85677 | 0.0064 |
| GDP_PER_CAPITA does not Granger Cause PRD | | 0.09699 | 0.7581 |
| RDE does not Granger Cause GDP_PER_CAPITA | 27 | 9.04547 | 0.0059 |
| GDP_PER_CAPITA does not Granger Cause RDE | | 0.07771 | 0.7827 |
| STJ does not Granger Cause GDP_PER_CAPITA | 27 | 3.93442 | 0.0584 |
| GDP_PER_CAPITA does not Granger Cause STJ | | 0.50925 | 0.4821 |
| THE does not Granger Cause GDP_PER_CAPITA | 27 | 4.39407 | 0.0463 |
| GDP_PER_CAPITA does not Granger Cause THE | | 0.07888 | 0.7811 |
| EDU does not Granger Cause GDP_PER_CAPITA | 27 | 2.43955 | 0.1309 |
| GDP_PER_CAPITA does not Granger Cause EDU | | 3.25965 | 0.0831 |

Pairwise Granger Causality Tests
Sample: 1990 2018 IF COUNTRY1 = 11

| Null Hypothesis: | Obs | F-Statistic | Prob. |
|---|---|---|---|
| PAN does not Granger Cause GDP_PER_CAPITA | 27 | 0.33389 | 0.7197 |
| GDP_PER_CAPITA does not Granger Cause PAN | | 0.11052 | 0.8959 |
| PAR does not Granger Cause GDP_PER_CAPITA | 27 | 0.33389 | 0.7197 |
| GDP_PER_CAPITA does not Granger Cause PAR | | 0.11052 | 0.8959 |
| PRD does not Granger Cause GDP_PER_CAPITA | 27 | 0.28754 | 0.7529 |
| GDP_PER_CAPITA does not Granger Cause PRD | | 0.09579 | 0.9090 |
| RDE does not Granger Cause GDP_PER_CAPITA | 27 | 0.31281 | 0.7346 |
| GDP_PER_CAPITA does not Granger Cause RDE | | 0.18231 | 0.8346 |
| STJ does not Granger Cause GDP_PER_CAPITA | 27 | 0.26862 | 0.7669 |
| GDP_PER_CAPITA does not Granger Cause STJ | | 1.67510 | 0.2103 |
| THE does not Granger Cause GDP_PER_CAPITA | 27 | 0.29089 | 0.7504 |
| GDP_PER_CAPITA does not Granger Cause THE | | 0.01289 | 0.9872 |
| EDU does not Granger Cause GDP_PER_CAPITA | 27 | 0.32060 | 0.7290 |
| GDP_PER_CAPITA does not Granger Cause EDU | | 0.02119 | 0.9791 |

Pairwise Granger Causality Tests
Sample: 1990 2018 IF COUNTRY1 = 12

| Null Hypothesis: | Obs | F-Statistic | Prob. |
|---|---|---|---|
| PAN does not Granger Cause GDP_PER_CAPITA | 26 | 0.89077 | 0.4638 |
| GDP_PER_CAPITA does not Granger Cause PAN | | 3.30388 | 0.0425 |
| PAR does not Granger Cause GDP_PER_CAPITA | 26 | 0.89077 | 0.4638 |
| GDP_PER_CAPITA does not Granger Cause PAR | | 3.30388 | 0.0425 |

**Table A1.** *Cont.*

| Null Hypothesis: | Obs | F-Statistic | Prob. |
|---|---|---|---|
| PRD does not Granger Cause GDP_PER_CAPITA | 26 | 0.45675 | 0.7157 |
| GDP_PER_CAPITA does not Granger Cause PRD | | 1.24709 | 0.3205 |
| RDE does not Granger Cause GDP_PER_CAPITA | 26 | 3.90725 | 0.0249 |
| GDP_PER_CAPITA does not Granger Cause RDE | | 0.69901 | 0.5642 |
| STJ does not Granger Cause GDP_PER_CAPITA | 26 | 0.74834 | 0.5367 |
| GDP_PER_CAPITA does not Granger Cause STJ | | 3.48123 | 0.0363 |
| THE does not Granger Cause GDP_PER_CAPITA | 26 | 1.98610 | 0.1503 |
| GDP_PER_CAPITA does not Granger Cause THE | | 1.64873 | 0.2117 |
| EDU does not Granger Cause GDP_PER_CAPITA | 26 | 0.59479 | 0.6261 |
| GDP_PER_CAPITA does not Granger Cause EDU | | 1.69819 | 0.2012 |

Pairwise Granger Causality Tests
Sample: 1990 2018 IF COUNTRY1 = 13

| Null Hypothesis: | Obs | F-Statistic | Prob. |
|---|---|---|---|
| PAN does not Granger Cause GDP_PER_CAPITA | 27 | 0.97908 | 0.3914 |
| GDP_PER_CAPITA does not Granger Cause PAN | | 0.66307 | 0.5253 |
| PAR does not Granger Cause GDP_PER_CAPITA | 27 | 0.97908 | 0.3914 |
| GDP_PER_CAPITA does not Granger Cause PAR | | 0.66307 | 0.5253 |
| PRD does not Granger Cause GDP_PER_CAPITA | 27 | 0.27895 | 0.7592 |
| GDP_PER_CAPITA does not Granger Cause PRD | | 0.45400 | 0.6409 |
| RDE does not Granger Cause GDP_PER_CAPITA | 27 | 0.91969 | 0.4134 |
| GDP_PER_CAPITA does not Granger Cause RDE | | 0.36466 | 0.6986 |
| STJ does not Granger Cause GDP_PER_CAPITA | 27 | 0.09426 | 0.9104 |
| GDP_PER_CAPITA does not Granger Cause STJ | | 5.11692 | 0.0150 |
| THE does not Granger Cause GDP_PER_CAPITA | 27 | 0.93801 | 0.4065 |
| GDP_PER_CAPITA does not Granger Cause THE | | 0.91330 | 0.4159 |
| EDU does not Granger Cause GDP_PER_CAPITA | 27 | 0.03313 | 0.9675 |
| GDP_PER_CAPITA does not Granger Cause EDU | | 0.17291 | 0.8423 |

Pairwise Granger Causality Tests
Sample: 1990 2018 IF COUNTRY1 = 14

| Null Hypothesis: | Obs | F-Statistic | Prob. |
|---|---|---|---|
| PAN does not Granger Cause GDP_PER_CAPITA | 27 | 1.32599 | 0.2859 |
| GDP_PER_CAPITA does not Granger Cause PAN | | 0.14147 | 0.8689 |
| PAR does not Granger Cause GDP_PER_CAPITA | 27 | 1.37618 | 0.2734 |
| GDP_PER_CAPITA does not Granger Cause PAR | | 1.62187 | 0.2203 |
| PRD does not Granger Cause GDP_PER_CAPITA | 27 | 0.33509 | 0.7189 |
| GDP_PER_CAPITA does not Granger Cause PRD | | 0.69119 | 0.5115 |
| RDE does not Granger Cause GDP_PER_CAPITA | 27 | 4.55742 | 0.0221 |
| GDP_PER_CAPITA does not Granger Cause RDE | | 0.67502 | 0.5194 |
| STJ does not Granger Cause GDP_PER_CAPITA | 27 | 0.15912 | 0.8539 |
| GDP_PER_CAPITA does not Granger Cause STJ | | 0.41448 | 0.6657 |
| THE does not Granger Cause GDP_PER_CAPITA | 27 | 1.78683 | 0.1910 |
| GDP_PER_CAPITA does not Granger Cause THE | | 0.26876 | 0.7668 |
| EDU does not Granger Cause GDP_PER_CAPITA | 27 | 3.61820 | 0.0438 |
| GDP_PER_CAPITA does not Granger Cause EDU | | 0.98913 | 0.3878 |

Pairwise Granger Causality Tests
Sample: 1990 2018 IF COUNTRY1 = 15

| Null Hypothesis: | Obs | F-Statistic | Prob. |
|---|---|---|---|
| PAN does not Granger Cause GDP_PER_CAPITA | 27 | 3.94050 | 0.0242 |
| GDP_PER_CAPITA does not Granger Cause PAN | | 0.95690 | 0.4332 |
| PAR does not Granger Cause GDP_PER_CAPITA | 27 | 0.58816 | 0.6302 |
| GDP_PER_CAPITA does not Granger Cause PAR | | 2.94843 | 0.0590 |

**Table A1.** *Cont.*

| Null Hypothesis: | Obs | F-Statistic | Prob. |
|---|---|---|---|
| PRD does not Granger Cause GDP_PER_CAPITA | 27 | 0.26065 | 0.8528 |
| GDP_PER_CAPITA does not Granger Cause PRD | | 0.80394 | 0.5071 |
| RDE does not Granger Cause GDP_PER_CAPITA | 27 | 1.05090 | 0.3930 |
| GDP_PER_CAPITA does not Granger Cause RDE | | 3.07919 | 0.0523 |
| STJ does not Granger Cause GDP_PER_CAPITA | 27 | 2.19014 | 0.1226 |
| GDP_PER_CAPITA does not Granger Cause STJ | | 0.18275 | 0.9068 |
| THE does not Granger Cause GDP_PER_CAPITA | 27 | 0.81335 | 0.5022 |
| GDP_PER_CAPITA does not Granger Cause THE | | 4.74172 | 0.0124 |
| EDU does not Granger Cause GDP_PER_CAPITA | 27 | 0.04317 | 0.9877 |
| GDP_PER_CAPITA does not Granger Cause EDU | | 1.17654 | 0.3449 |

Pairwise Granger Causality Tests
Sample: 1990 2018 IF COUNTRY1 = 16
Lags: 2

| Null Hypothesis: | Obs | F-Statistic | Prob. |
|---|---|---|---|
| PAN does not Granger Cause GDP_PER_CAPITA | 27 | 2.56142 | 0.1000 |
| GDP_PER_CAPITA does not Granger Cause PAN | | 6.73806 | 0.0052 |
| PAR does not Granger Cause GDP_PER_CAPITA | 27 | 2.40498 | 0.1136 |
| GDP_PER_CAPITA does not Granger Cause PAR | | 0.14059 | 0.8696 |
| PRD does not Granger Cause GDP_PER_CAPITA | 27 | 0.85992 | 0.4369 |
| GDP_PER_CAPITA does not Granger Cause PRD | | 0.29628 | 0.7465 |
| RDE does not Granger Cause GDP_PER_CAPITA | 27 | 1.02146 | 0.3765 |
| GDP_PER_CAPITA does not Granger Cause RDE | | 0.13856 | 0.8714 |
| STJ does not Granger Cause GDP_PER_CAPITA | 27 | 9.11119 | 0.0013 |
| GDP_PER_CAPITA does not Granger Cause STJ | | 19.9675 | 1.E-05 |
| THE does not Granger Cause GDP_PER_CAPITA | 27 | 0.35056 | 0.7082 |
| GDP_PER_CAPITA does not Granger Cause THE | | 11.8021 | 0.0003 |
| EDU does not Granger Cause GDP_PER_CAPITA | 27 | 2.68070 | 0.0908 |
| GDP_PER_CAPITA does not Granger Cause EDU | | 2.94423 | 0.0736 |

Pairwise Granger Causality Tests
Sample: 1990 2018 IF COUNTRY1 = 17

| Null Hypothesis: | Obs | F-Statistic | Prob. |
|---|---|---|---|
| PAN does not Granger Cause GDP_PER_CAPITA | 27 | 0.13404 | 0.8753 |
| GDP_PER_CAPITA does not Granger Cause PAN | | 0.34464 | 0.7122 |
| PAR does not Granger Cause GDP_PER_CAPITA | 27 | 1.79948 | 0.1889 |
| GDP_PER_CAPITA does not Granger Cause PAR | | 9.57892 | 0.0010 |
| PRD does not Granger Cause GDP_PER_CAPITA | 27 | 1.56537 | 0.2314 |
| GDP_PER_CAPITA does not Granger Cause PRD | | 1.02108 | 0.3767 |
| RDE does not Granger Cause GDP_PER_CAPITA | 27 | 0.27437 | 0.7626 |
| GDP_PER_CAPITA does not Granger Cause RDE | | 1.95295 | 0.1657 |
| STJ does not Granger Cause GDP_PER_CAPITA | 27 | 0.98285 | 0.3901 |
| GDP_PER_CAPITA does not Granger Cause STJ | | 0.50109 | 0.6126 |
| THE does not Granger Cause GDP_PER_CAPITA | 27 | 1.00865 | 0.3810 |
| GDP_PER_CAPITA does not Granger Cause THE | | 1.46224 | 0.2534 |
| EDU does not Granger Cause GDP_PER_CAPITA | 27 | 0.59774 | 0.5587 |
| GDP_PER_CAPITA does not Granger Cause EDU | | 0.34051 | 0.7151 |

Pairwise Granger Causality Tests
Sample: 1990 2018 IF COUNTRY1 = 18

| Null Hypothesis: | Obs | F-Statistic | Prob. |
|---|---|---|---|
| PAN does not Granger Cause GDP_PER_CAPITA | 27 | 4.52198 | 0.0226 |
| GDP_PER_CAPITA does not Granger Cause PAN | | 1.67446 | 0.2104 |
| PAR does not Granger Cause GDP_PER_CAPITA | 27 | 1.14622 | 0.3361 |
| GDP_PER_CAPITA does not Granger Cause PAR | | 0.80727 | 0.4589 |

**Table A1.** *Cont.*

| Null Hypothesis: | Obs | F-Statistic | Prob. |
|---|---|---|---|
| PRD does not Granger Cause GDP_PER_CAPITA | 27 | 1.54357 | 0.2359 |
| GDP_PER_CAPITA does not Granger Cause PRD | | 1.21407 | 0.3161 |
| RDE does not Granger Cause GDP_PER_CAPITA | 27 | 0.76778 | 0.4761 |
| GDP_PER_CAPITA does not Granger Cause RDE | | 2.69477 | 0.0898 |
| STJ does not Granger Cause GDP_PER_CAPITA | 27 | 6.39219 | 0.0065 |
| GDP_PER_CAPITA does not Granger Cause STJ | | 4.38578 | 0.0249 |
| THE does not Granger Cause GDP_PER_CAPITA | 27 | 8.52687 | 0.0018 |
| GDP_PER_CAPITA does not Granger Cause THE | | 0.56740 | 0.5751 |
| EDU does not Granger Cause GDP_PER_CAPITA | 27 | 5.85179 | 0.0092 |
| GDP_PER_CAPITA does not Granger Cause EDU | | 0.02474 | 0.9756 |

Pairwise Granger Causality Tests
Sample: 1990 2018 IF COUNTRY1 = 19

| Null Hypothesis: | Obs | F-Statistic | Prob. |
|---|---|---|---|
| PAN does not Granger Cause GDP_PER_CAPITA | 27 | 0.25761 | 0.6162 |
| GDP_PER_CAPITA does not Granger Cause PAN | | 7.66646 | 0.0104 |
| PAR does not Granger Cause GDP_PER_CAPITA | 27 | 3.59486 | 0.0696 |
| GDP_PER_CAPITA does not Granger Cause PAR | | 6.67297 | 0.0160 |
| PRD does not Granger Cause GDP_PER_CAPITA | 27 | 0.04494 | 0.8338 |
| GDP_PER_CAPITA does not Granger Cause PRD | | 2.59188 | 0.1200 |
| RDE does not Granger Cause GDP_PER_CAPITA | 27 | 9.86050 | 0.0043 |
| GDP_PER_CAPITA does not Granger Cause RDE | | 1.16887 | 0.2900 |
| STJ does not Granger Cause GDP_PER_CAPITA | 27 | 2.20958 | 0.1497 |
| GDP_PER_CAPITA does not Granger Cause STJ | | 16.5676 | 0.0004 |
| THE does not Granger Cause GDP_PER_CAPITA | 27 | 0.10476 | 0.7489 |
| GDP_PER_CAPITA does not Granger Cause THE | | 15.6075 | 0.0006 |
| EDU does not Granger Cause GDP_PER_CAPITA | 27 | 0.00675 | 0.9352 |
| GDP_PER_CAPITA does not Granger Cause EDU | | 7.53871 | 0.0110 |

Pairwise Granger Causality Tests
Sample: 1990 2018 IF COUNTRY1 = 19

| Null Hypothesis: | Obs | F-Statistic | Prob. |
|---|---|---|---|
| PAN does not Granger Cause GDP_PER_CAPITA | 27 | 0.25761 | 0.6162 |
| GDP_PER_CAPITA does not Granger Cause PAN | | 7.66646 | 0.0104 |
| PAR does not Granger Cause GDP_PER_CAPITA | 27 | 3.59486 | 0.0696 |
| GDP_PER_CAPITA does not Granger Cause PAR | | 6.67297 | 0.0160 |
| PRD does not Granger Cause GDP_PER_CAPITA | 27 | 0.04494 | 0.8338 |
| GDP_PER_CAPITA does not Granger Cause PRD | | 2.59188 | 0.1200 |
| RDE does not Granger Cause GDP_PER_CAPITA | 27 | 9.86050 | 0.0043 |
| GDP_PER_CAPITA does not Granger Cause RDE | | 1.16887 | 0.2900 |
| STJ does not Granger Cause GDP_PER_CAPITA | 27 | 2.20958 | 0.1497 |
| GDP_PER_CAPITA does not Granger Cause STJ | | 16.5676 | 0.0004 |
| THE does not Granger Cause GDP_PER_CAPITA | 27 | 0.10476 | 0.7489 |
| GDP_PER_CAPITA does not Granger Cause THE | | 15.6075 | 0.0006 |
| EDU does not Granger Cause GDP_PER_CAPITA | 27 | 0.00675 | 0.9352 |
| GDP_PER_CAPITA does not Granger Cause EDU | | 7.53871 | 0.0110 |

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
