# Peer review of "Causality between Technological Innovation and Economic Growth: Evidence from the Economies of Developing Countries"

_sustainability, doi:10.3390/su14063586_

Round 1

Reviewer 1 Report

Thank you very much for the opportunity to read "Causality between technological innovation and

economic growth: evidence from the economies of developing countries".

The topic of the research is very important.

However, I have a few suggestions.

References in the text to bibliographic references do not strictly follow the rules of the journal in question.

The writing of the paper shows a relatively careless way on the part of the authors.

I recommend the following to the authors to better identify the elements of their own scientific contribution.

In particular, to specify which is the part through which the paper brings superior elements in relation to other researchers.

The bibliography should be extended with some papers (8-10 titles) published in prestigious WoS indexed Journals.

Author Response

Response to the Editor and Reviewers

Original Manuscript ID: 1629162

Dear Respected Editor and Reviewers,

We are truly grateful for the comments on our Research Article titled: " Causality between technological innovation and economic growth: evidence from the economies of developing countries". These comments are highly insightful, which can enable us to further improve the quality of our manuscript. According to these comments, we have made careful modifications to the original manuscript. Revision portions are marked in yellow here in this note and included in the revised manuscript.

Our point-by-point response to each reviewer is listed in the following section.

Comments and Suggestions for Authors

Thank you very much for the opportunity to read "Causality between technological innovation and economic growth: evidence from the economies of developing countries".

The topic of the research is very important.

However, I have a few suggestions.

Review Report#1

Concern # 1: References in the text to bibliographic references do not strictly follow the rules of the journal in question.

Author response: The question has been addressed. 

Author action: The amendment was made in accordance with the rules of the relevant journal.

Concern # 2: The writing of the paper shows a relatively careless way on the part of the authors.

Author response: The question has been addressed. 

Author action: The writing has been modified as much as possible

Concern # 3:I recommend the following to the authors to better identify the elements of their own scientific contribution.

Author response: The question has been addressed. Page 7 

Author action: Our study contributes to the literature in the following ways. First, to our knowledge, this is the first study to find a systematic relationship between technological innovation and growth in the economy of developing countries. There are many studies that talk about the relationship between innovation in general and economic growth, and our study focused on technological innovation, because it is considered one of the most important types of innovation in addition to being one of the basic and important activities of contemporary institutions, as the main reason for the existence of institutions is to provide distinguished products and services In order for it to survive and grow, it must adapt to changes in the external environment and find the necessary methods and processes to enable it to offer all new or improved products and services to achieve superiority over competitors, especially in the developing economy. Second, we document this through the results of the study, so we find that the technological innovation index represented in the percentage of spending on education is expected to have a positive impact on countries, while the results were completely different. It had a negative and moral impact of 1% and this does not fit with the various theoretical analyzes that considered spending on education as a driver of economic growth. This is because developing countries still need to spend more on education infrastructure for innovation to deliver its expected results.

Concern # 4: The writing of the paper shows a relatively careless way on the part of the authors.

Author response: The question has been addressed. Page 7 

Author action: Therefore, the results were consistent with what was reported in previous studies (abdelaoui. et al.,(2020), Lomachynska & Podgorna, (2018), Pece et al (2015), Solomon et al (2011)., . with what was mentioned in the studies by K. Tuna et al( 2015)).

Concern # 5: The bibliography should be extended with some papers (8-10 titles) published in prestigious WoS indexed Journals.

Author response: The question has been addressed. 

Author action: The list of references has been expanded with some of the research papers published in the prestigious Web of Science Indexed Journals.

Reviewer 2 Report

First of all, I want to thank you for the effort to write this manuscript but I have some important considerations that in my point of view are of high importance:

Comments on title, abstract, references:

1. The title matches the abstract well. It conveys the main idea of the article and also is somewhat interesting.

2.      The abstract is well designed regarding the aim, and brief method.
3.      The references are relevant and somewhat recent. To the best of my knowledge, they correctly cited and no major references are missing.

Comments on introduction:
The study context is acceptable. However, the research question need to be addressed clearly. In addition, it could be better by add some contents to develop the importance for the subject of interest. More discussion is needed to highlight the significant of the subject. Moreover, I could suggest also to add one sentence for the presenting the structure sections of the paper and their content which is very useful for reader.

Comments on Theoretical Background:
Authors stated that (Among the most important studies that focused on this aspect, we find the works of (Freimane & Bāliņa, 2016; Maradana et al., 2017( used research and development as a measure of innovative activities [15-19]). However, I would criticize them about the citation format, which should be numbered in square brackets [1], [2] ... etc. according to MDPI templates. Kindly apply the same for the whole manuscript as there were many similar cases.

I would also advise you to try to follow the MDPI Instructions for Authors for the Sustainability journal, https://www.mdpi.com/journal/sustainability/instructions, especially in the General consideration section where it is specifically specified about Research manuscript sections: Introduction, Materials and Methods, Results, Discussion, Conclusions (which are always at the end :) (For example: Kindly refer to Conclusion and limitation which you have included future research in it).

Comments on conclusions and policy recommendations:
Conclusions must also be revised. In particular, they should discuss practical and academicians implications and even policy makers and management as well as future lines of research. As it stands now, they fail to extract all the juice of this work. I would recommend the authors to add more contributions and recommendation; the presentation of future work in the area need to be more addressed and detailed. (In general, the section of conclusion, limitations and future recommendations need to be revised and improved. I recommend that you slightly rearrange/rename them to match the structure with the MDPI requirements).

I hope that all these questions and comments can help to improve your manuscript.

Author Response

Response to the Editor and Reviewers

Original Manuscript ID: 1629162

Dear Respected Editor and Reviewers,

We are truly grateful for the comments on our Research Article titled: " Causality between technological innovation and economic growth: evidence from the economies of developing countries". These comments are highly insightful, which can enable us to further improve the quality of our manuscript. According to these comments, we have made careful modifications to the original manuscript. Revision portions are marked in yellow here in this note and included in the revised manuscript.

Our point-by-point response to each reviewer is listed in the following section.

Comments and Suggestions

First of all, I want to thank you for the effort to write this manuscript but I have some important considerations that in my point of view are of high importance:

Review Report#2

Comments on title, abstract, references:

  1. The title matches the abstract well. It conveys the main idea of the article and also is somewhat interesting.
  2. The abstract is well designed regarding the aim, and brief method.
  3. The references are relevant and somewhat recent. To the best of my knowledge, they correctly cited and no major references are missing

Comments on introduction:

Concern1 #:The study context is acceptable. However, the research question need to be addressed clearly. In addition, it could be better by add some contents to develop the importance for the subject of interest. More discussion is needed to highlight the significant of the subject. Moreover, I could suggest also to add one sentence for the presenting the structure sections of the paper and their content which is very useful for reader.

Author response: The question has been addressed. Page 6,7,8 

Author action: The significance of our study is that Since the second half of the twentieth century, the world has moved towards expanding the use of the Internet and its applications with continuous progress in practical research, to witness breakthroughs in the world of innovation and technology, which have been reflected positively and gradually in the sectors of the economy, industry, health, and agriculture. Researchers addressed this phenomenon through observation and analysis, and some described it as the fourth industrial revolution. In this context, the importance of research comes: It “leads innovation to economic growth and thus the economies of developing countries through technology and innovation”, to advanced economies and therefore developing countries study the strategic importance of innovation and technology in providing unconventional solutions to global challenges, especially with increasing demand On food, energy and water, and how to promote global urbanization. It then attempts to redraw the mental image of rich countries as not monopolizing the capabilities of innovation, but it needs the forces of minds from all over the world to provide solutions to common global challenges and exchange and distribute international burdens.

Hence, Economic progress is no longer associated with the possession of natural resources or material possibilities، as it is linked to the content of knowledge، technology، quality and innovation. Japan is a country without resources, but with attention to human resources and economic innovation, Japan was able to be among the most important economies in the world and achieved the highest rates of gross domestic product, so the research problem lies in the fact that countries that suffer from weak knowledge and technology content cannot upgrade Its economy and economic growth unlike developed countries that possess advanced technology and have a long history of innovation; It can achieve great economic growth. Growth in developing countries faces serious constraints, partly due to the lack of innovation, which is at the same time the reason for these countries to remain underdeveloped. These barriers arise from inappropriate business activity, governance, and poor education. In such cases, innovation itself is encouraged to deal with difficult situations [23].

The following is an explanation of the importance of this study:

  • It is considered an important applied research in studying the relationship between technological innovation and economic growth.
  • Intensifying the research and development process for the purpose of changing the traditional structures in the developing country, and thus providing new goods that would improve the financial conditions and consequently the economic growth of that country.
  • Transition from the traditional economy to the innovative one by acquiring various skills that enable countries to improve their financial performance.
  • The researcher expects, through this study, to motivate researchers to conduct more research in the field of technological innovation, especially in the relationship between innovation, sustainable development, and competitiveness.

Based on the foregoing, the study problem is limited to answering the following study question:

Is there a causal relationship between economic growth and technological innovation in the group of developing countries understudy?

Our study is based on the main hypothesis that:

- There is a long-term causal relationship between economic growth and technological innovation in both directions for the group of developing countries under study.

 The remainder of this paper is organized as follows. The Theoretical Background follows in Section 2. In Section 3, the materials description, and variables are included used in the analysis. Causality tests were performed at Granger, followed by a co-integration and error correction model (ECM). Section 4 presents the Estimating and Analyzing Results. In Section 5, conclusions are drawn. Finally, Conclusions in Section 6,

 Comments on Theoretical Background:

Concern # 1:  Authors stated that (Among the most important studies that focused on this aspect, we find the works of (Freimane & Bāliņa, 2016; Maradana et al., 2017( used research and development as a measure of innovative activities [15-19]). However, I would criticize them about the citation format, which should be numbered in square brackets [1], [2] ... etc. according to MDPI templates. Kindly apply the same for the whole manuscript as there were many similar cases.

Author response: The question has been addressed. Page 3 

Author action: Citation writing is standardized according to MDPI templates in all this Paper

Freiman & Bāliņa, Maradana et al.,

Concern # 2: I would also advise you to try to follow the MDPI Instructions for Authors for the Sustainability journal, https://www.mdpi.com/journal/sustainability/instructions, especially in the General consideration section where it is specifically specified about Research manuscript sections: Introduction, Materials and Methods, Results, Discussion, Conclusions (which are always at the end :) (For example: Kindly refer to Conclusion and limitation which you have included future research in it).

Author response: The question has been addressed.  Page 31

Author action: Follow MDPI's instructions

  1. Conclusions and Recommendations

Despite this contribution, as with any research, it comes with some limitations. First, given the method of data collection chosen, most developing countries imply a lack of data and a fallacy in that data. This might be considered a limitation of the study. Secondly, most developing countries are still importing technology from developed countries that may not suit their environment, and this leads to not reaping the fruits of technological innovation expected to be obtained. Finally, our study found that the economic problem today is based on the abundance of information and not the traditional scarce resources, because of technological innovation. Moreover, economic growth has become the decisive element in all aspects of economic activity, and knowledge has become the basis for any economic or social growth, and through that, the world has shifted from research and collision in order to Sources of scarce resources to search and clash in order to control as many knowledge sources as possible.

 Therefore, future research should look at how to direct economic resources towards knowledge industries in a manner equivalent to the volume of resources directed towards investments in the sectors of construction, tourism, sports, and entertainment.  and the need to search for ways to support scientific research and researchers in the field of knowledge technologies and increase the volume of spending on scientific research so that it constitutes a good percentage of Gross National Product, which has a positive impact on the country's national economy. Based on previous analyzes and research conclusions, the following suggestions have been articulated to provide the basis for improving levels of technological innovation in developing countries. For this and based on the results, in this study, the following recommendations can be made: (1) For the policymakers Strategy should be adopted to improve high-tech, export, rather than export raw and primary goods. (2), the government should provide appropriate funding for R&D performed in the public sector, especially in the higher education sector. (3(Encouraging the policy of economic openness for developing countries, by taking reform measures at various levels in order to benefit from trade openness to the outside world in the field of innovation, and thus support economic growth .(4) Working to create an environment conducive to innovation in developing countries, by expanding spending on research and development, as well as by protecting intellectual property rights.

Comments on conclusions and policy recommendations:

Conclusions must also be revised. In particular, they should discuss practical and academicians implications and even policy makers and management as well as future lines of research. As it stands now, they fail to extract all the juice of this work. I would recommend the authors to add more contributions and recommendation; the presentation of future work in the area need to be more addressed and detailed. (In general, the section of conclusion, limitations and future recommendations need to be revised and improved. I recommend that you slightly rearrange/rename them to match the structure with the MDPI requirements).

Author response: The question has been addressed. Page 30,31 

Author action:  6. Conclusions and Recommendations

In this paper, we examined the relationship between technological innovation and economic growth over the period 1990-2018. The study was conducted in twenty developing countries. Comparison between countries, considering the growth rate of the country's per capita GDP (the growth of the country's economy is measured as a percentage increase in the gross national product per capita), and the independent variables were education expenditures and patents by residents and non-residents, research and development expenditures, and researchers in research and development activities High-tech, export, and scientific and technical journal articles. The study concluded that the variables relied upon have a positive impact on economic growth. The results showed the significant positive impact of research and spending development, scientific and technical journals, and high-tech exports on per capita GDP growth, while PAN, PAR, and RDE had little effect on GDP per capita, however, the data show Mexico has the highest average PAR, while Tunisia has the lowest one. While Argentina has the highest average PAN, while Tunisia has the lowest one while Hungary has the highest average RDE, China and Indonesia have the lowest averages, Hungary has the highest PRD average while the lowest averages are in China, Indonesia, and the Philippines. As for THE, and DUE, Hungary has the highest average. The lowest value of these two variables is in China. Poland has the highest STJ score while the lowest is in the Philippines, while China has the largest average per capita GDP, while the lowest in Argentina. In short, we conclude that a country with a high rate of investment in R&D, high-tech exports, patent rights, and science and technology journal articles will be highly developed, and that country's economy will be very innovative. Hence, the level and structure of innovation should not be overlooked as it plays a fundamental role in stimulating economic growth.

Despite this contribution, as with any research, it comes with some limitations. First, given the method of data collection chosen, most developing countries imply a lack of data and a fallacy in that data. This might be considered a limitation of the study. Secondly, most developing countries are still importing technology from developed countries that may not suit their environment, and this leads to not reaping the fruits of technological innovation expected to be obtained. Finally, our study found that the economic problem today is based on the abundance of information and not the traditional scarce resources, because of technological innovation. Moreover, economic growth has become the decisive element in all aspects of economic activity, and knowledge has become the basis for any economic or social growth, and through that, the world has shifted from research and collision in order to Sources of scarce resources to search and clash in order to control as many knowledge sources as possible.

 Therefore, future research should look at how to direct economic resources towards knowledge industries in a manner equivalent to the volume of resources directed towards investments in the sectors of construction, tourism, sports, and entertainment.  and the need to search for ways to support scientific research and researchers in the field of knowledge technologies and increase the volume of spending on scientific research so that it constitutes a good percentage of Gross National Product, which has a positive impact on the country's national economy. Based on previous analyzes and research conclusions, the following suggestions have been articulated to provide the basis for improving levels of technological innovation in developing countries. For this and based on the results, in this study, the following recommendations can be made: (1) for the policymakers Strategy should be adopted to improve high-tech, export, rather than export raw and primary goods. (2), the government should provide appropriate funding for R&D performed in the public sector, especially in the higher education sector(3(Encouraging the policy of economic openness for developing countries, by taking reform measures at various levels in order to benefit from trade openness to the outside world in the field of innovation, and thus support economic growth. (4) Working to create an environment conducive to innovation in developing countries, by expanding spending on research and development, as well as by protecting intellectual property rights.

Reviewer 3 Report

Interesting paper on growth and the causality relation between technological innovation and economic growth.

Need revision of the abstract for clarity and the coherence with the overall structure of the paper and its objectives and results or conclusions.

Is education a proxy for technological innovation? It would be important to distinguish the factors that promote growth and distinguish which are from technology or technological innovation and which are coming from other factors, such as demographics, education, natural resources. This lack of conceptual clarity cast some doubts on the methodology, and the results as well.

The methods should be better explained and introduced. The data of several figures and tables could be improved in terms of clarity.

Another point is the lack of comments and discussion of the data presented (see some examples in the file I have attached.) Table 3, for instance, is not duly discussed. (By the way the reference of the table or figure is not always correct)

The names of countries in figures 2 and following.

The variables should be conceptually discussed (Patents fields by Non-residents (PAN), Researchers Scholars in research and development activities (RDE): calculated per 1,000 population; Researcher’s development and expenditure (PRD): Measured as a proportion of actual GDP, High technology exports (HTE): Measured as a proportion of the real domestic output and Scientific and technical journals (STG): Measured by one thousand people.)

Another point is the lack of situating the paper with the traditional literature on innovation, technological change and growth. The solow residual that represents “technological change” is not even discussed.

You could look at authors such as Nathan Rosenbert, Paul Romer, M. Trajtenberg, Solow, Arrow (for learning by doing), Barro, Sala-i-Martin, Maddison, among others. And econometric references should be reinforced (see Granger-cause and other recent papers on causality).

The overall style and sentences must be revised. I suggest reviewing frases like this one and many others for style, clarity and sometimes grammar.

The discussion - before the conclusion - should be extended. Some results are interesting and should be compared to the literature on economic growth and on technological change and some institutions.

Author Response

Response to the Editor and Reviewers

Original Manuscript ID: 1629162

Dear Respected Editor and Reviewers,                                                     

We are truly grateful for the comments on our Research Article titled: " Causality between technological innovation and economic growth: evidence from the economies of developing countries". These comments are highly insightful, which can enable us to further improve the quality of our manuscript. According to these comments, we have made careful modifications to the original manuscript. Revision portions are marked in yellow here in this note and included in the revised manuscript.

Our point-by-point response to each reviewer is listed in the following section.

Comments

Review Report#3

Interesting paper on growth and the causality relation between technological innovation and economic growth.

Concern # 1:  Need revision of the abstract for clarity and coherence with the overall structure of the paper and its objectives and results or conclusions.

Author response: The question has been addressed.  Pages 1

Author action:  Abstract: Economic growth is a tool for measuring the development and progress of countries, and technological innovation is one of the factors affecting economic growth and contributes to the development and modernization of production methods. Therefore, technological innovation is the main driver of economic growth and human progress. Thus, spending on innovation, research and development, and investment in innovation supports competition and progress. Accordingly, sustainable economic growth is achieved. This ensures the preservation of resources for future generations and the achievement of economic and social growth. Moreover, a sustainable educational level of the workforce will be ensured, investment in research, creation of new products, and investor access to stock markets will be ensured through the development of the public and private sectors and the improvement of people's living conditions. Our study aimed to measure the impact of technological innovation on economic growth in developing countries during the period (1990-2018). To this end, the Error Correction Model (ECM) method has been applied. The results showed that the variables are unstable in the level and stable after taking the first difference. Co-integration was also tested using the Error Correction Model (ECM), and Granger's causality test for the direction of causation. The test results showed that an increase in technological innovation indicators (such as spending on education, number of patents for residents and non-residents, R&D expenditures, number of researchers in R&D, high-tech exports, and scientific and technical research papers.) leads to an increase in economic growth in the short term. and the long-run, with a long-run and two-way causal relationship between technological innovation and GDP, and short-run causation going from technological innovation to GDP. The study concluded also, that technological innovation has a direct impact on the sustainability of the country's economic growth, which is why it is so important to adopt strong policies to encourage international investors to allocate capital for development in the developing country and encourage more research and development.

Concern # 2:  - Is education a proxy for technological innovation? It would be important to distinguish the factors that promote growth and distinguish which are from technology or technological innovation and which are coming from other factors, such as demographics, education, natural resources. This lack of conceptual clarity cast some doubts on the methodology, and the results as well.

Author response: The question has been addressed. 

Author action:  Pages 2,3,4

Economic growth is the continuous increases in real income in the long term, and the force increases in income are considered economic growth. and economic development a structural and radical change in most of the structures of the national economy, unlike growth, which focuses only on the change in the volume of goods and services obtained by the individual represented by an increase in his average income.

Hence, economic growth is an increase in the economy's ability to produce goods and services during a specified period. It refers to the long-term expansion of the productive potential of the economy to meet the needs of individuals in society. The sustainable economic growth of the country has a positive impact on the national income and the level of employment, which leads to more standards of living. There are many factors that affect economic growth: (1) the amount of physical capital; The availability of more auxiliary tools in production processes leads to more output of goods and services, and accordingly, the output of the individual, from the accumulation of capital, became noticeable, to the extent that it was considered at one time, that the physical capital is the only source of economic growth in general. Investment opportunities that were not presented before, it is possible for this society to achieve an increase in its production capacity by increasing its balance of real capital. It must reveal, sooner or later, the decrease in the return on capital according to the decrease in its marginal productivity with every increase in the quantity used in the production process. Along the line, one of the most prominent examples of this is the impact of physical capital on the economic growth of the United States. During the current century, that is, despite the huge amounts of marginal capital used in that stage of the development of the American economy, the ratio of output to capital has remained in proportion and in proportion to the declining trend and did not deteriorate in proportion. Extremely important, is that investment opportunities have expanded at the same speed as investment in capital goods. (2) Human resources: are one of the most important factors leading to increased economic growth; the quantity and quality of human resources contribute directly to the economy. The quality of human resources depends on a set of characteristics, the most important of which is their ability to innovate, education, training, and skills. In the event of a shortage of skilled human resources, this will hinder economic growth. (3) Natural resources: are among the factors affecting the economic growth of a country. What is significant, and includes all the natural resources that appear on the surface of the earth or within it, such as plants on land, and water resources. The natural resources within the earth include gas, oil, and minerals. Natural resources differ between countries based on their environmental and climatic conditions. (4) Social and political factors: They are the factors that aim to play an important role in the economic growth of countries. Traditions, customs and beliefs constitute social factors, while government participation in policy development and implementation constitutes political factors. (5) Technological development: One of the important and influencing factors in economic growth, and it includes the application of a set of productive techniques and scientific methods, and technology is defined as the nature and quality of technical tools, dependent on the use of a certain percentage of the workforce. Technology is defined as “a set of knowledge, experiences, and practices.” Technology and the interrelationships between the sub-systems of work, whereby its application and adoption contributes to satisfying actual or expected economic and social needs[4, 5].

In the same context, (6) innovation is one of the factors that affect economic growth; innovation can be defined as “the activity that produces new or significantly improved goods (products or services), processes, marketing methods, or business organizations[6]. This definition focuses on forms of innovation in It may be embodied either in a new or improved product, and it can also be defined as “the successful commercial exploitation of new ideas and includes all scientific, technological, organizational and financial activities that lead to the provision of everything new (or improvement) of a product or service[7, 8]. Innovation also refers to “the successful exploitation of new ideas”[9]. According to (Sarvan, Atalay, 2013), innovation can be embodied in the following manifestations: - Creating new products or qualitative improvements in existing products. - Carrying out a new industrial process. - Opening a new market. - Developing new sources of raw materials or other new inputs. - New forms of industrial organizations[10].

Here are several types of innovation, and they are usually classified according to the following criteria: Classification of innovation according to the output criterion and includes two types: product innovation and process innovation. Innovation is also classified according to the market perception criterion, and this classification includes two basic types: continuous innovation and intermittent or discontinuous innovation. Innovation is also classified according to the criterion of the size of change (according to Degree): According to this criterion, innovation is divided into two types: radical innovation, and improvement innovation (gradual - partial). Alternatively, a production method to the process of achieving it and embodying it in a tangible form.

Finally, classification of innovation according to the criterion of specialization into managerial innovation, marketing innovation, and technological innovation.” According to "Garcia (2014) “Technological innovation is a set of technical, industrial and commercial stages that lead to the launch of manufactured and commercial products and the use of new technical processes[11].” The following figure 1. Shows the types of technological innovation.

Figure 1. Types of technological innovation

Source: Prepared by the authors based on previous studies

From the previous figure 1. it is clear that technological innovation consists of two types: product innovation, which is either introducing a new product or improving an existing product, and the second type is process innovation, which consists of designing a new process or improving an existing process. The innovative process, where countries today depend on the use of modern technology to remove many of the barriers that make the country more open and developed in terms of speed of completion of work and keeping pace with the times, by focusing on the research and development function in a way that allows it to keep pace with these developments and challenges and adapt to them. As countries cannot maintain their level of performance, regardless of their capabilities or capabilities, if they rely on traditional methods in the era of the technological revolution. For this, countries must rely on technological innovation, which is one of the most important pillars of the development of countries where it can reach the required level of performance efficiently and effectively.

Concern # 3: - The methods should be better explained and introduced. The data of several figures and tables could be improved in terms of clarity.

Author response: The question has been addressed.  Page 9-12

Author action: 3. Research Model

Economic methodology

Aghion and Howitt proposed a model for the variables that affect innovation: the technical multiplier, total employment committed to innovation, intermediate product output, and volume of final and medium goods produced. We demonstrate that the driving factor behind economic growth is the creation of technology developments. This outcome relates to the financial framework, that is, the demand that allows the creator to finance and, in some way, the possibility of excluding the enterprise from the business. Therefore, when speaking of innovation as a transformation in the production process, as formulated by Schumpeter, Cobb-Douglas will be called the product advantage with constant returns to scale, namely[39, 40]:

Yt = AK t α L t 1− α     (1)

Where Yt is production, Kt is capital, Lt is labor, A is the technological coefficient, and α and 1 −α is, respectively, The share of capital and labor in production. To assess the evolution of Total Factor Productivity TFP, [41]should be considered, taking into account the contribution of capital to increasing demand, calculated by the change in the percentage of capital that doubles its market share; And follow the same theory, the proportional change due to labor is the rise in the amount of production compounded by the share. The growth rate of the TFP is defined by variables other than labor and resources. Such considerations include the effective utilization of energy, technical developments, and innovation in R&D, patents, and exports of high-tech goods. As normal, the TPF is obtained by taking logarithms in (1), and is given as follows:

TFP= gQ − SK gK − SLgL         (2)

Where the growth rate of production is GQ, SK is the share of capital in the industry, gK is the rate of growth of capital, SL is the share of labor in the product, and gL is the rate of growth of labor.

Standard Methodology

In standard studies, Panel Data refers to multi-directional data mostly that includes measurements over time and contains data for multiple phenomena over time and for the same economic units. Panel Data models have become increasingly popular in the field of applied studies due to their high ability to study human behavior compared to time-series models or cross-section models, and Panel Data has become increasingly rich and available in all developed and developing countries alike. The World Bank is tasked with helping to design many of the surveys for Panel Data. Panel Data is done by adding a sample of a particular unit over time to other units within a group, thus providing multiple observations for each unit of the sample.

Panel Data has several characteristics that distinguish it from data for cross-sections or time series. It works to control individual variance that may lead to biased results. It also provides an expansion of the sample size used by researchers, increasing degrees of freedom and reducing interdependence between explanatory variables, thus helping to improve the efficiency of estimates. Statistical, as well as cross-sectional data. On the other hand, the panel data allows researchers to analyze a number of important economic questions that cannot be studied using time series or cross-sections alone. Moreover, the Panel Data allows for the construction or testing of more complex models; This is done by utilizing information at the temporal dynamic level and at the individual level of the panels being studied. It also provides the possibility of generating more accurate predictions for units. In general, the regression can be represented as follows[42, 43]:

(3)

Whereas:

()       Study variables vectors.

Panel Data cross-sectional directions (i =1,..,. N),

(N) represents the number of units (people, companies, industries, countries ... etc.),

(t=1,….,T) time direction.

Stability tests, Time series are divided according to the stability characteristic into Stationary series: Series whose levels change with time without changing the average in them during a relatively long period of time, i.e., where there is no general trend towards in which there is a general trend towards either increase or decrease (does not contain a unit root).

Un Stationary series: These are the series whose mean is constantly changing, increasing or decreasing (containing a unit root)[44].  using panel data, which is defined as cross-sectional data, are measured at certain time intervals. The main benefit of using them is to increase prediction accuracy by increasing the number of views by associating the number of cross-sectional views with the number of time periods[44].

Economic data are often characterized by the presence of structural changes that affect the degree of the indifference of time series, so determining the degree of inactivity is important before testing integration and causation relationships, as this requires data instability and integration of the same degree. If the variable is not fixed at the level, while it was fixed at the level of the first differences, then it is an integrated variable of the first degree. A stable time series will be (static) if it has the following characteristics:

- The stability of their average values ​​over time, i.e.

 E (xt) = μ

- The stability of the variability of their values ​​over time, i.e. V (Xt) = E (Xt-μ) 2 = σ2     

- The covariance between two values ​​of the same variable depends on the time gap between the two values ​​and not on the actual value of time i.e.

 cove (x t, xt+k)=E[( Xt-μ) .( Xt-k-μ)]= γΚ

That is, the time series is considered to have a stable covariance if its means and covariances are constant [45]over time to determine if the variables () are stable or not, the Augmented Dickey-Fuller (1981) ADF test is performed. To perform the ADF test, we use the following equation

+     (4)

The instability hypothesis is rejected when the parameter is negative (d) and significant. If the variables are stable and integrated of the first degree, we move to the next step, to find out whether the variables are jointly integrated and that there is a long-term equilibrium relationship between the variables. After that, the following two hypotheses are tested:

(The variable Y does not remain stationary = contains a unit root) Ho: β<0

(The variable Y rests at its level = integral of degree zero) Ha: β=0

The null hypothesis is rejected if the calculated t value is greater than the tabulated or critical value of t (in absolute value suggested by MacKinnon (Mackinnon 1991). Nevertheless, if the variable is not static at the level while it is static at the level of the first differences, then it is an integrated variable of the first degree (1). In general, the series xt is integrated from the degree d if it is static at the level of the differences d, so it contains several d is a unit root [46-48]. After conducting unit root tests for the variables under study, it is proven that the variables are characterized as Integral of the first degree (1), it is possible to conduct joint integration between them. The basis of the co-integration method is that two or more non-static variables can be co-integrated (they have a long-run equilibrium relationship) if one of them is in regression over the other and the residuals themselves are stationary. As Engle & Granger pointed out, even if the time series (individually) are not stationary, their linear structures can be stationary, because the equilibrium forces tend to hold these time series together in the long run. When this happens, the variables can be considered co-integrated. Hence, the error-correcting vocabulary is created to consider the short-term deviation from the long-run equilibrium relationship resulting from the co-integration [49-51].

Granger's causality test; Granger has demonstrated how to introduce the traditional method for causation testing when using the Error Correction Model (ECM). By using, the error correction model derived from the cross-integration. The ECM also makes it possible to distinguish between the long run and the short run. Where the F and T-tests of the first difference variables deceleration indicate causation in the short term, while the error correction factor indicates causation in the long - run[52, 53].

Where: ∆ The first difference,  error correction limit If the estimates of the two parameters () are statistically significant, then this indicates the existence of a long-term causal relationship in two directions from Yt to Xt and vice versa. Whereas if only  is significant, then this means that there is a one-way causal relationship from Xt to Yt (this implies that Xt leads Yt to long-run equilibrium).

The hysteresis values ​​​​∆Yt and ∆Xt represent explanatory variables in the model, and indicate the causal relationship in the short term. If the parameters of ​​∆Yt are the previous equation number 5 is significant, it means that Y causes X[54].

On the other hand, if any integrative vector is not reached for a long-term relationship between the study variables, we can detect the causal relationship between variables in the short term through the Granger Speaker in the Multiple Frame in the Self-Region (VAR) model:

Since: Yt-1,1 -Xt-1 are the slowed study variables, m the lag period, μi, λi are the parameters of the slowed variables Yt, the random limit.

Thus, the Granger causality test is used to ascertain the extent to which there is a feedback, or an interrelationship between two variables. In this study, a model of the causal relationship between technological innovation and economic growth will be estimated using the Granger method. The results were as shown in Table (No. 3).

This research seeks to test the relationship between technological innovation and economic growth achieved by these countries, which are, respectively: Argentina, Algeria, Brazil, Bulgaria, Chile, China, Egypt, India, Indonesia, Iran, Mexico, Morocco, Peru, Philippines, Poland, Romania, Sri Lanka, Thailand, Tunisia, and Turkey. The research was conducted during the period 1990-2018. The main variables used to measure innovation were Expenditure on education (DUE): The percent of education expenditure in GDP for each of the developing countries.

Concern # 4: Another point is the lack of comments and discussion of the data presented (see some examples in the file I have attached.) Table 3, for instance, is not duly discussed. (By the way the reference of the table or figure is not always correct)

Author response: The question has been addressed. 

Author action Table 3. displays the results of the ADF test. Using the unit root ADF test at each of the individual countries, using the statistical program (EViews).The results of the stability tests in the plane (in a model with a single constant and direction, with a single constant, without a single constant and direction) indicate that all-time series is unstable in the plane where the corresponding probability of these tests in most of the models was greater than the significance limit (0.05) or (0.1). As for stability tests in the first differences, the results indicate that the remaining time series are all stable in the first differences in all models, that is, it is (1) l, where the corresponding probability of these tests was less than the significance limit (0.05 or 0.1). The stability of time series at the level and in the first differences means there is the possibility of a co-integration relationship between these time series, as shown in Table 4.

Table 4. The degree of integration of the variables of the model under study.

Variables

PAR

PAN

RDE

PRD

THE

STJ

EDU

GDP

Degree of integration

I (1)

I (1)

I (1)

I (1)

I (1)

I (1)

I (1)

I (1)

Prepared by researchers based on Table 3

Concern # 5: The names of countries in figures 2 and following.

 Author response: The question has been addressed. 

Author action: The names of the twenty countries used in the study have been clarified. Page 12 to 13

Concern # 6: The variables should be conceptually discussed (Patents fields by Non-residents (PAN), Researchers Scholars in research and development activities (RDE): calculated per 1,000 population; Researcher’s development and expenditure (PRD): Measured as a proportion of actual GDP, High technology exports (HTE): Measured as a proportion of the real domestic output and Scientific and technical journals (STG): Measured by one thousand people.).

Author response: The question has been addressed.  Page  6;7

Author action: (DUE):  Public expenditure on education as a percentage of total government expenditure is the total public expenditure (current and capital) on education, expressed as a percentage of GDP in any year. Public expenditure on education includes items of government expenditure on educational institutions (public and private), education administration as well as transfers/subsidies to private entities (students/families, other private entities).

(RDE): Patent applications are worldwide patent applications filed through the Patent Cooperation Treaty (PCT) or with a national patent office to register exclusive ownership of innovation - whether it is a product or a process that involves a new way of making something or offers a new technical solution to a problem. A patent provides protection for the invention for the benefit of the patent owner for a limited period, generally up to 20 years.

(PRD): Gross domestic expenditures on research and development (R&D), are expressed as a percent of GDP. They include both capital and current expenditures in the four main sectors: Business enterprise, Government, Higher education, and Private non-profit. R&D covers basic research, applied research, and experimental development.

(HTE): High-technology exports are products with high R&D intensity, such as aerospace, computers, pharmaceuticals, scientific instruments, and electrical machinery.

(STG): Scientific and technical journal articles refer to the number of scientific and engineering articles published in the following fields: physics, biology, chemistry, mathematics, clinical medicine, biomedical research, engineering and technology, and earth and space sciences.

Concern # 7: - Another point is the lack of situating the paper with the traditional literature on innovation, technological change, and growth. The Solow residual that represents “technological change” is not even discussed.

Author response: The question has been addressed. page  (18)

Author action:. The Solow residual is a number describing empirical productivity growth in an economy from year to year and decade to decade. Robert Solow, the Nobel Memorial Prize in Economic Sciences-winning economist, defined rising productivity as rising output with constant capital and labor input. It is a "residual" because it is the part of the growth that is not accounted for by measures of capital accumulation or increased labor input. Increased physical throughput – i.e. environmental resources – is specifically excluded from the calculation; thus, some portion of the residual can be ascribed to increased physical throughput. The example used is for the intracapital substitution of aluminum fixtures for steel during which the inputs do not alter. This differs in almost every other economic circumstance in which there are many other variables. The Solow Residual is procyclical and measures of it are now called the rate of growth of multifactor productivity or total factor productivity, According to the “Solow surplus” model, the unexplained portion of economic growth, except labor and capital increase, is technological development. The convergence hypothesis, which is one of the main implications of the Solow model, is based on the assumption that technological change is external and constant between countries. Accordingly, per capita output levels of countries will approach each other, and the development differences will automatically disappear in the long term. Technological change is one of the most important challenges facing countries for its strategic role in achieving outstanding performance, maintaining its competitive advantage in the markets, and its sustainability, survival, and success in the fields of work. Technological change is a more comprehensive concept than development, growth, and progress. Technological change is what leads to development Technological development can be defined as a set of activities related to examining, evaluating, and implementing an idea or goal for the purpose of moving from the research mental level to the production level, and includes developing processes for technical capabilities, performance, design, engineering model, and manufacturability. While technological growth: means a continuous increase in technology over time. While technological progress: is the change in the art of production used, leading to an increase in productivity, provided that the ratio of capital and labor use remains constant.

While, Both Ricardo and Adam Smith emphasize that openness will enhance specialization and thus countries will specialize in the production of goods and services that have advantages and export these goods and services, on the other hand, countries that do not have these advantages will import from those countries and specialize in other types of goods and services, and as a result, resources are allocated optimally, when we find that the theory of internal growth indicates that developing countries will benefit from the transfer of advanced technology through a policy of trade openness, this technology that can be exploited in productive processes and thus achieve a large production that is directly reflected in economic growth.

The neoclassical growth models derived from Solo's 1957 model consider a technological change to be exogenous and suggest that trade policies do not, therefore, affect economic growth. However, new economic growth theories assume that technological change is an endogenous variable.

Through the foregoing, technological change can be defined as “the use of innovation or creativity outputs for the purpose of bringing about a partial or total change in the production process, or the product that aims to support competitiveness and therefore continuous modification in it to achieve continuity and growth.” Thus, modern growth theories emerged, which are termed internal growth theories, with the contributions of (Romer, 1986) and (Lucas 1988), and the theory of internal growth focused on the internal impact of technological change, research and development, human capital, and their impact on the production function[36, 37]. In-house designed technological change generates sustainable economic growth, assuming constant returns to innovative research, in terms of human capital used in research and development (R&D). Internal growth models provide an appropriate framework for examining important issues related to the role of technological change in the process of economic growth, as well as design, research and development efficiency and innovation policies. "Barro"focused on infrastructure and public expenditures, and others focused on economic openness and its role in economic growth[38].

Concern # 8: - You could look at authors such as Nathan Rosenbert, Paul Romer, M. Trajtenberg, Solow, Arrow (for learning by doing), Barro, Sala-i-Martin, Maddison, among others. And econometric references should be reinforced (see Granger-cause and other recent papers on causality).

Author response: The question has been addressed.

Author action References have been used and added as much as possible.

Concern # 9: - the overall style and sentences must be revised. I suggest reviewing frases like this one and many others for style, clarity and sometimes grammar.

Author response: The question has been addressed.   

Author action: Writing improved as much as possible

Concern # 10: - The discussion - before the conclusion - should be extended. Some results are interesting and should be compared to the literature on economic growth and on technological change and some institutions.

Author response: The question has been addressed.

Author action: 5. Discussion

This study aimed to measure the Causality between technological innovation (Expenditure on education (EDU); Patent fields by residents (PAR); Patents fields by Non-residents (PAN); Researchers Scholars in research and development activities (RDE); Researcher’s development and expenditure (PRD); High technology exports (HTE); and Scientific and technical journals (STG)). As Independent Variables, and economic growth on economic growth in 20 developing countries, for the period (1990-2018). Using the economic measurement of the panel data, the study concluded that the economic growth of the countries under study is linked to a causality relationship with some technological Innovation indicators. Through the above, we find that Mexico has the highest average PAR, followed by Hungary, while Tunisia has the lowest average. Hungary has the highest average RDE, while China and Indonesia have the lowest averages. Hungary has the highest average PRD, while China, Indonesia, and the Philippines have the lowest averages. Hungary has the highest average for THE and DUE. , while China has the lowest average of these two variables. While Poland has the highest STJ average, the Philippines has the lowest. China recorded the highest average per capita GDP, while Argentina recorded the lowest.

The results also indicate to of the stability tests in the plane (in a model with a single constant and direction, with a single constant, without a single constant and direction) that all-time series is unstable in the plane where the corresponding probability of these tests in most of the models was greater than the significance limit (0.05) or (0.1). As for stability tests in the first differences, the results indicate that the remaining time series are all stable in the first differences in all models, that is, it is (1) l, where the corresponding probability of these tests was less than the significance limit (0.05 or 0.1). The stability of time series at the level and in the first differences means there is the possibility of a co-integration relationship between these time series.

Also, The results of the estimation showed the statistical significance of four variables (EDU, PRD, STJ, THE). Where the estimation results showed there is a negative and significant effect of Expenditure education (EDU). This means that an increase in spending on education leads to a decrease in economic growth, this does not fit with various theoretical analyzes that have considered education spending as a driver of economic growth. This is due to, developing countries still needing more spending on education infrastructure so that innovation brings its expected results. While there is a positive and significant effect of Research and development expenditure (PRD) on economic growth. In addition, scientific and technical journal articles (STJ) have positive and significant impact on economic growth. As for the variable of High technology exports (THE), which has a positive and noticeable impact. This means that the variables of technological innovation (Researcher’s development and expenditure, Scientific and technical journals, and High technology exports) had an important role in influencing the economic growth in the countries under study, and this is consistent with the various theoretical analyzes that considered these variables as a driving factor for economic growth.

In addition, we found that as for the variables (Patents fields by Non-residents (PAN), Patent fields by residents (PAR), and Researchers Scholars in research and development activities (RDE)), all estimation attempts were unsuccessful in finding an important and significant relationship showing the extent to which the impact of technological innovation contributes to economic growth. Where it turned out that, there is a weak and non-significant negative relationship for the variable Patents fields by Non-residents (PAN). As for the variable Patent fields by residents, its effect was positive and insignificant, and its contribution was very weak, as the increase in this percentage would not lead to high economic growth. As for the variable of researchers in research and development activities, its relationship to economic growth was negative, strong, and not significant. Nevertheless, here there is no effect because it is not significant in the model. Therefore, the results were consistent with what was reported in previous studies (abdelaoui. et al.,(2020), Lomachynska & Podgorna, (2018), Pece et al (2015), Solomon et al (2011)., . with what was mentioned in the studies by K. Tuna et al( 2015)).

Round 2

Reviewer 1 Report

Improvements in the work are significant. 

Author Response

Response to the Editor and Reviewers

Original Manuscript ID: 1629162

Dear Respected Editor and Reviewers,

We are truly grateful for the comments on our Research Article titled: " Causality between technological innovation and economic growth: evidence from the economies of developing countries". These comments are highly insightful, which can enable us to further improve the quality of our manuscript. According to these comments, we have made careful modifications to the original manuscript. Revision portions are marked in yellow here in this note and included in the revised manuscript.

Review Report# 1

Comments and Suggestions for Authors

Improvements in the work are significant. 

Author action: Thank you very much.

Reviewer 3 Report

 I think that the paper has been much improved. I still have some questions. First, are there no changes in the references? I would suggest references on innovation and growth theory. For example Richard Nelson, R. Winter, etc. 

The abstract has been improved. The text is overall more coherent. I still consider that the discussion of growth theory and innovation theory could be improved. A lot of resources can be found in SPRU research center of the university of Sussex, for example. 

Author Response

Response to the Editor and Reviewers

Original Manuscript ID: 1629162

Dear Respected Editor and Reviewers,

We are truly grateful for the comments on our Research Article titled: " Causality between technological innovation and economic growth: evidence from the economies of developing countries". These comments are highly insightful, which can enable us to further improve the quality of our manuscript. According to these comments, we have made careful modifications to the original manuscript. Revision portions are marked in yellow here in this note and included in the revised manuscript.

Our point-by-point response is listed in the following section.

Review Report# 3

Comments and Suggestions for Authors

Concern1 #: I think that the paper has been much improved. I still have some questions. First, are there no changes in the references? I would suggest references on innovation and growth theory. For example Richard Nelson, R. Winter, etc. 

  Author action:  The question has been addressed. References numbers

 [ 4 to12,24,36,37to40,59,60,67,68,69,78,79,80]

Author response: The economist Joseph Schumpeter considered that innovation is one of the productive functions and emphasized that entrepreneurs are able to achieve these innovations and thus entrepreneurship plays a fundamental role in economic growth[30].

While Paul Romer's model of endogenous growth distinguishes between inputs and outputs. His knowledge takes the form of a number of ideas (designs) that are embodied in the form of a number of (technical) inputs, which in turn are embodied in the form of final goods and services. Hence, Paul Romer's model linked the sector of production of ideas and designs (research and development), the sector of input production (the sector of production of intermediate goods), the sector of capital production (which is just a mixture of inputs) and the sector of production of goods and services.[22, 41]. Hence, it can be said - according to Romer's model - that designs constitute the output of the knowledge economy, while the inputs that are used in the production of capital and in the final goods production sector represent the impact of the knowledge economy on the knowledge-based economy. Thus, this relationship between these sectors is logical to govern - in principle - the logic of designing and building knowledge standards, knowledge economy standards, and knowledge-based economy standards. Romer concludes that growth is often driven by the accumulation of non-competitive inputs (intermediate inputs), but they are partially enumerated, and by competitive inputs embodied in human capital, not by the size of the labor force or the size of the population[22]. Thus, the transition from a product economy to a knowledge economy has some consequences, including providing an opportunity to increase returns, such as what happened in the industries software sector, as well as creating the opportunity to benefit freely, by taking advantage of knowledge outputs[42].

In the same context ,some studies, Aghion and Howitt 1998, Chu2013, Jinli Zeng, 2002, indicate that capital accumulation (both physical and human) and innovation should not be considered as causal factors differentiate, but are manifestations of a single process. On the one hand, capital is used in the innovation process and in new technology applications resulting from research and development activities. Hence, long-term growth depends on both capital accumulation and innovation. On the other hand, new technologies create new economic opportunities for investment in physical and human capital[43-45]. Nelson has indicated that knowledge takes the first priority compared to the traditional factors of production, material and financial. and unlike land, labor, and capital, which were highlighted by traditional economists as final factors of production, knowledge, and ideas are infinite goods and help to obtain increased benefits, the new economists suppose linked to the theory of superior growth Creativity emanating through the system[46]. Nelson also emphasized that the level of innovative activity in a country is determined by the level of interaction of specialized [47]institutions among them[48]. Hence, a review of these different theories confirms that technological progress appears in them as a supportive factor for productivity growth and thus achieving long-term economic growth[49]. Expenditure on scientific research, technical development, education, and rehabilitation of human capital is one of the most important tools supporting innovation[39].

 Hence, most innovation studies focused on developing solutions to technology problems. Researchers have tried to show how the organization can develop technological solutions to the problems they face, where technology is seen as solutions to problems[50]. In addition, the results of many quantitative research confirm that the development of technological capabilities is a prerequisite for reducing the difference in economic development between countries and thus achieving the so-called catch-up growth in developed countries. (Catch-up)[51]. Means reducing the difference in the level of income per capita. Many countries, such as Japan, South Korea, and others have also achieved this. The economist Kim interpreted the economic development in South Korea on the basis of the development of its technological capabilities, which is known as the ability to effectively use technical knowledge to imitate, invest, localize and modify the existing technology. Technology capabilities are also a necessary condition for achieving technology transfer and settlement[52]. Whereas, innovation potential describes a country's ability to produce and market innovative technology over the long term[53]. The financial and scientific resources necessary for innovation and the results of scientific research are the most important factors that affect the innovative potential of a country[54]. Also, human capital, infrastructure, and foreign n trade are among the most important factors affecting this country's ability to absorb new technology. Achieving development based on innovation and thus achieving economic growth.

Concern # 2:  The abstract has been improved. The text is overall more coherent. I still consider that the discussion of growth theory and innovation theory could be improved. A lot of resources can be found in SPRU research center of the university of Sussex, for example.

Author response: The question has been addressed. pages 8,9,10,11 

Author action: The following references have been used and added.

 ( 22,30,39,41,42,43,44,45,46,47,48,49,51,51,52,53,54)